# Profile of predominant gram-negative pathogenic bacteria in River Sosiani and wastewater systems in Eldoret Town, Uasin Gishu County, Kenya

Sharon Auma,[1] James E. Barasa,[2] Caroline Kosgei,[3] Naomi Bisem,[1] Salinah Rono,[4] Richard Korir[5]

**ABSTRACT** Gram-negative pathogenic bacteria play a significant role in spreading infections, with some strains exhibiting resistance to multiple antibiotics. Aquatic and wastewater systems, which receive effluents from various sources, contain pathogenic bacteria, chemicals, and antibiotic contaminants. This study investigated the bacterial load and antibiotic resistance profiles of gram-negative bacteria in water samples from wastewater systems and River Sosiani in Eldoret town, Kenya. Water samples were aseptically processed using standard microbiological techniques, followed by antibiotic susceptibility testing using the disc diffusion method. Data were coded and analyzed using Microsoft Excel and ANOVA. The highest bacterial count was detected at Kipkaren bridge (296) and the lowest at Kipkenyo boundary effluent (21). The study identified pathogenic gram-negative bacteria with varying frequencies: *Escherichia coli* (30.3%), *Enterobacter aerogenes* (20.9%), *Klebsiella pneumoniae* (10.3%), *Salmonella enteritidis* (8.7%), *Citrobacter freundii* (7.8%), *Yersinia enterocolitica* (5.6%), *Proteus vulgaris* (5.6%), *Proteus mirabilis* (5.1%), *Pseudomonas aeruginosa* (4.4%), and *Klebsiella oxytoca* (2.7%). Antibiotic susceptibility tests revealed that all isolates were susceptible to ciprofloxacin, doxycycline, gentamicin, and chloramphenicol, with high multi-antibiotic resistance indices recorded in *E. coli* (60%), *E. aerogenes* (33.3%), *C. freundii* (30%), *K. oxytoca* (30%), *K. pneumoniae* (25%), *P. mirabilis* (25%), *P. vulgaris* (16.7%), *P. aeruginosa* (12.5%), and *Y. enterocolitica* (12.5%) but not in *S. enteritidis* (0%). The study demonstrates rivers and wastewater systems as the critical reservoirs of pathogenic bacteria, exhibiting varying levels of multi-antibiotic resistance index. This poses threats of multi-drug resistant infections outbreak from the environment to public health, hence prompting the need for quick remedial action.

**IMPORTANCE** The study highlights the critical role of wastewater and aquatic systems as reservoirs for pathogenic gram-negative bacteria, which play a key role in the spread of infections. The findings reveal an alarming presence of various antibiotic-resistant bacteria, with particularly high multi-antibiotic resistance index in several species to commonly used antibiotics. These findings demonstrate the urgent need for enhanced wastewater treatment practices and the regular monitoring of water sources to curb the spread of waterborne diseases and safeguard public health.

**KEYWORDS** aquatic systems, wastewater, pathogenic bacteria, antibiotic resistance, multi-antibiotic resistance index

Gram-negative pathogenic bacteria from aquatic environments are the key drivers in the rise and spread of various infectious bacterial diseases (1–3). Aquatic ecosystems often receive influxes, including wastewater from fish farms and hatcheries, medical facilities, mines, farms, quarries, logging, and sites of deforestation, which invariably

Address correspondence to Sharon Auma, aumasharon97@gmail.com.

The authors declare no conflict of interest.

contain different bacteria (4–6). Some of these bacteria are discharged directly or indirectly through diffusion, leakage, and surface runoff, especially during flooding. Apart from being sources of pathogenic bacteria, these dynamic systems also carry chemicals and contaminants that induce resistance in bacteria to antibiotics (7–9). Water bodies provide water for domestic use, livestock watering points, shipping and transport, irrigation, and industrial and recreational activities, which indirectly contribute to antibiotic-resistant bacteria in the aquatic environments (10, 11). Such bacteria proliferate, develop, and spread multi-drug-resistant infections to humans, livestock, and aquatic biota (12–14). Therefore, the application of antibiotics to treat and manage infections caused by such bacteria, which is often a common approach to controlling pathogenic infections, is rendered ineffective. Naturally, bacterial pathogenicity and toxicity increase in these water sources and wastewater systems, making water unsafe to potential users (15, 16). Many studies report different confirmed cases of increased resistance to multiple antibiotics by species of *Escherichia coli, Enterobacter, Pseudomonas, Proteus, Salmonella, Yersinia,* and other enteric bacteria isolates from hospital wastewater (17), East Africa (18), and many other parts of the world (19, 20).

Antimicrobial resistance (AMR) is the ability of microbes to hinder the effective actions of antibiotics, thus making these antibiotics inefficient to inhibit the growth of different microorganisms such as bacteria, fungi, and viruses (21). Antibiotic resistance among different species is a major global threat to public health, which contributed between 1.27 and 4.95 million deaths globally in 2019 (22–24). Furthermore, antibiotic-resistant infections present a heavy burden to healthcare systems in many countries (25), especially in the tropical and subtropical regions, since they require alternative and more innovative treatment strategies, substantially increasing the cost of healthcare in the world (26, 27). Despite antimicrobial resistance in bacteria being driven by several factors (28–30), antibiotic residues in the environment are still the potential factors that foster the development of bacterial resistance (31–33). Pathogenic bacteria develop and acquire defense mechanisms to evade the effects of antibiotics, hence becoming resistant (34, 35). These bacteria express either plasmid-mediated resistance or chromosomal DNA-mediated mechanisms (36), including limiting drug uptake by bacteria, inactivating and hydrolyzing drugs, modifying drug targets, inducing genetic, enzymatic, and metabolic changes in the hosts that neutralize the efficacy of antibiotics, and potentially allowing extrusion of antibiotics through different efflux pumps (37). However, many studies have emphasized the need for regular monitoring of microbial quality of various water systems to support proper mitigation for better human health, but the AMR issue has not been fully resolved (20, 38, 39).

As part of the One Health approach to address the widespread global challenge of antibiotic resistance among patients at hospitals and the community, as well as in agriculture, livestock, and fisheries, the Government of Kenya developed the National Action Plan (NAP) on antimicrobial resistance (40), consistent with the Global Action Plan of 2015. The objectives of the National Action Plan (NAP) include enhancing awareness and understanding of antimicrobial resistance (AMR) through effective communication, education, and training, as well as strengthening the knowledge and evidence base through surveillance and research (40). In line with these goals, this study was conducted to isolate and identify pathogenic gram-negative bacteria from water systems and determine their antibiotic susceptibility and resistance profiles against commonly used antibiotics in medical facilities. The study was designed to support ongoing efforts of local authorities by providing evidence-based data to guide informed interventions and contribute to the development of a suitable policy framework to address antimicrobial resistance. The findings from this study are expected to raise awareness and concern among citizens, public health officials, environmental health personnel, and authorities about the increasing prevalence of pathogenic bacteria species in aquatic ecosystems. This prompts the need for precautionary measures, effective prevention strategies, and enhanced treatment options for patients by utilizing new and more effective antibiotics.

## MATERIALS AND METHODS

### Study area

The study was conducted along River Sosiani, which flows through Eldoret town in Uasin Gishu County, Kenya and several wastewater treatment plants serving the town. The sampling locations spanned from Kipkaren Bridge (upstream) at 0°30'52"N, 35°15'22"E, to Pioneer Bridge (center) at 0°30'46"N, 35°16'26"E, and finally to Outspan-Nairobi Bridge (downstream) at 0°29'53"N, 35°18'09"E. In addition, the study included influent and effluent points from the Huruma quarry (influent: 0°31'21"N, 35°14'20"E; effluent: 0°31'52"N, 35°14'12"E) and Kipkenyo Boundary sewage treatment plants (influent: 0°31'33"N, 35°12'44"E; effluent: 0°32'32"N, 35°12'49"E). Influent site represents the entry point of raw, untreated wastewater and other wastes into sewage treatment plants, whereas effluent site represents the point where the final treated wastes of the wastewater treatment plant are discharged into the nearby water body. Additional sites, including the waste discharge points from Moi Teaching and Referral Hospital (MTRH) (0°30′44″N, 35°16′50″E) and Eldoret Prison (0°31′34″N, 35°17′15″E), discharge pretreated waste into the municipal sewer lines, which are eventually directed to the Huruma quarry and Kipkenyo WWTPs for further treatment.

The study area, covering 42.2 km² within Eldoret, is heavily influenced by dense human settlements, agricultural activities, and small-scale industrial operations, all contributing to the discharge of wastewater and pollutants into River Sosiani. These activities create conditions conducive to the emergence of antibiotic-resistant bacteria in the river's ecosystem, highlighting the challenges of waste management and treatment in the region. Quantum geographic information system (QGIS) version 3.34.3 (41) was used to design the study map area according to the specific geographical location coordinates of the sampling sites within Eldoret city, (0°30'54"N 35°16'19"E), Uasin Gishu, Kenya and the nine sampling sites used in the study are shown in Fig. 1.

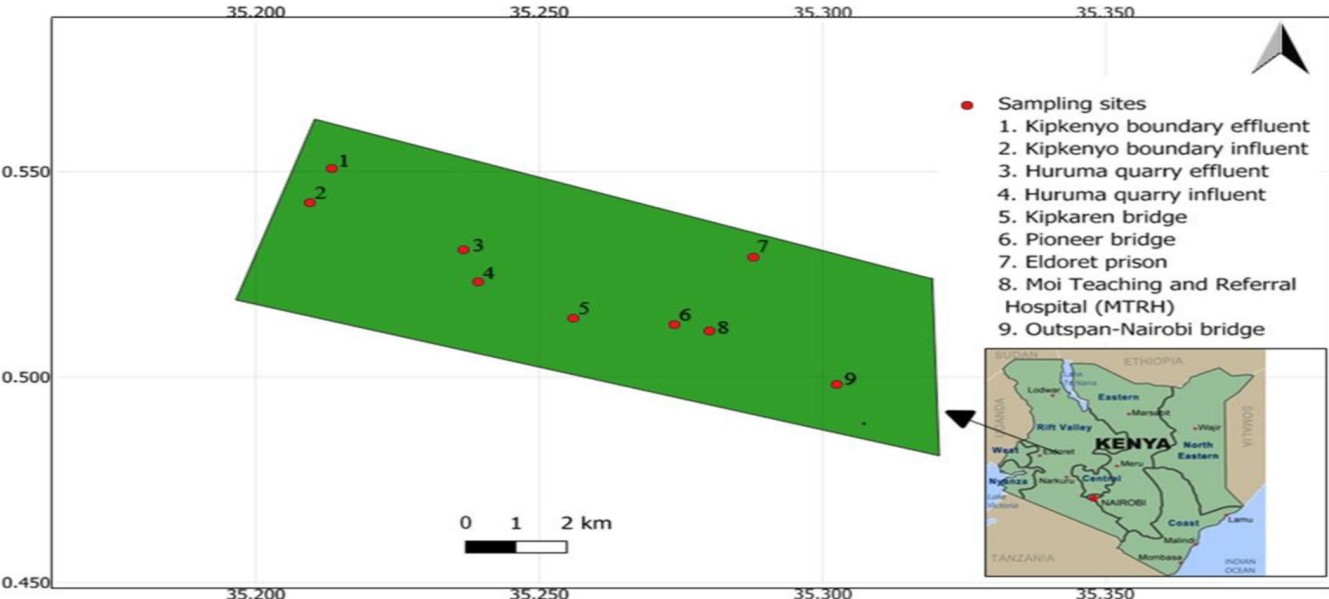

**FIG 1** Map of the study area showing locations of the nine sites sampled in the current study: Kipkenyo boundary influent, Kipkenyo boundary effluent, Huruma quarry influent, and Huruma quarry effluent sites are the wastewater treatment plants managed by Eldoret Water and Sanitation company (ELDOWAS), Kenya. Kipkaren Bridge, Pioneer Bridge, and Outspan -Nairobi Bridge are located along River Sosiani. Moi Teaching and Referral Hospital (MTRH) and Eldoret Prison are the waste disposal screen sites pretreated by ELDOWAS.

## Study design

Purposive sampling method was used to select key areas to be included in the study, whereas a cross-sectional study design was used to delineate specific sampling points for wastewater and river water samples. This included three River Sosiani bridges, Huruma quarry, and Kipkenyo boundary influent and effluent points, waste discharge points of Eldoret Prison and Moi Teaching and Referral Hospital.

## Sample collection and processing

A total of nine (9) water samples were collected aseptically in triplicate from selected sampling sites using sterilized 100 mL plastic containers. The samples were immediately transported in a cool box to the central water testing and microbiology laboratory of the Eldoret Water and Sanitation Company (ELDOWAS) in Eldoret, Kenya, for general bacterial culturing and wastewater analysis. Upon arrival, the samples were refrigerated at 4°C until processing. For turbidity-based dilution, highly concentrated samples from wastewater discharge points were diluted by taking 0.5 mL of the sample and adding it to 100 mL of sterile distilled water. For less concentrated samples from River Sosiani, 1 mL of the sample was added to 100 mL of distilled water. The diluted samples were then inoculated onto Endo agar plates and incubated at 37°C for 24 h to allow bacterial growth.

Sample dilutions for the isolation of fecal coliforms were prepared in physiological saline. Using sterilized forceps, a sterile grid filter paper was placed over the porous plate of the filtration apparatus, ensuring that the grid side faced up and the unit was securely locked into place. A 100 mL sample was passed through the filter under partial vacuum until the entire volume was drained into a volumetric flask. Once the filtration was complete, the filtration unit was unlocked, and the Buchner funnel, along with the soaked membrane filter, was removed using sterilized forceps. The membrane filter, which contained the trapped bacteria, was carefully placed onto solidified Endo agar plates in a rotational manner to ensure even distribution of bacteria across the surface. The plates were then incubated at 44°C for 24 h. The resulting colonies were examined for the presence of fecal coliforms, as indicated by their characteristic color and growth pattern on the Endo agar. The total coliforms and fecal coliforms were quantitatively counted using a digital colony counter and recorded as colony-forming units per 100 mL (CFU). For diluted samples, the CFU count was multiplied by the dilution factor to obtain the accurate number of bacterial colonies. Plates with colony counts exceeding 10,000 CFU were recorded as "too numerous to count" (TNTC).

## Culture and biochemical tests

The gram-negative bacteria coliforms were sub-cultured on Eosin Methylene Blue agar media (EMB) (Himedia) in the laboratory of Biological Sciences, University of Eldoret, Kenya. Distinct identification of these isolated pure bacterial colonies was based on the colony morphology and pigmentation on EMB plates. Gram-negative bacteria were initially identified based on their shape using Gram staining (42), followed by subspecies differentiation through biochemical tests, with some modifications to standard protocols (43, 44). Five biochemical tests were employed, based on the morphological and cultural characteristics observed on eosin methylene blue (EMB) agar plates. The catalase test was used to detect bacteria capable of producing the catalase enzyme, which decomposes hydrogen peroxide (45). The citrate test identified bacteria that could utilize citrate as a sole energy source (46), whereas the oxidase test, performed using the filter paper method, detected the presence of cytochrome c oxidase (47). The indole test confirmed bacteria capable of producing tryptophanase (48), and the triple sugar iron (TSI) test identified bacteria capable of fermenting glucose, lactose, and sucrose, as well as producing hydrogen sulfide gas (44).

## Antibiotic susceptibility testing in bacteria

Antibiotic susceptibility testing (AST) was performed using Mueller-Hinton Agar (MHA) prepared and adjusted to a 0.5 MacFarland turbidity standard (49). Pure bacterial isolates were suspended in normal saline and then spread evenly onto the prepared MHA plates. Using sterilized forceps, antibiotic discs were placed onto each of the MHA plates. The plates were incubated at 37°C for 24–48 h to assess bacterial susceptibility and resistance to the selected antibiotics. The number of plates prepared was based on the number of bacterial isolates from each sampling site. Antibiotic susceptibility and resistance profiles of different bacterial isolates were determined using the Kirby-Bauer disc diffusion method on Mueller-Hinton Agar (MHA) (50, 51), following the most recent guidelines from the Clinical and Laboratory Standards Institute (52, 53). The bacteria isolates were tested against 20 standard antibiotic discs, which included penicillin (P, 30 µg), ampicillin (AMP, 10 µg), levofloxacin (L, 2 µg), streptomycin (S, 10 µg), sulfame-thoxazole-trimethoprim (SXT, 20 µg), kanamycin (K, 30 µg), oxacillin (OX, 1 µg), cotrimox-azole (COT, 25 µg), doxycycline (DO, 30 µg), erythromycin (E, 15 µg), vancomycin (VA, 30 µg), amoxicillin-clavulanic acid (AMC, 10/20 µg), ceftazidime (CAZ, 30 µg), minocycline (MI, 30 µg), tetracycline (TET, 30 µg), azithromycin (AZT, 15 µg), gentamicin (GE, 10 µg), chloramphenicol (CH, 50 µg), metronidazole (MET, 5 µg), and ciprofloxacin (C, 30 µg). The antibiotic susceptibility testing was done in triplicates, and the obtained mean values and standard deviations for the zone of inhibition were recorded in millimeters (mm). Quality control was ensured by strictly adhering to standardized guidelines for sample collection, reagent preparation, and aseptic techniques, including autoclaving equipment and media, and proper storage of antibiotics, reagents, and bacterial cultures at recommended temperatures.

## Data analysis

All data recorded were coded and exported into EpiData software and analyzed using Microsoft Excel. The percentage of each bacterial isolate identified from a specific sampling location was calculated using the formula: (a/b) × 100%, where $a$ refers to the number of specific bacterial isolates obtained at each site, and $b$ is the total number of that particular bacterial isolate detected across all nine samples. One-way analysis of variance (ANOVA) was performed to assess variations in the number of different resistant bacterial isolates detected from the various samples. Antibiotic susceptibility testing results were interpreted based on the diameter of the zone of inhibition measured in millimeters (mm). The antibiogram patterns of bacterial isolates were classified as either susceptible (S), intermediate susceptible (IS), or resistant (R) (54, 55). Descriptive statistics, including averages and standard deviations, were used to summarize the data, with the following interpretation of the inhibition zones: 0 mm = resistance (R), 1–10 mm = intermediate susceptibility (IS), and >10 mm = susceptibility (S) (56). The Multi-Antibiotic Resistance Index (MARI) for each bacterial isolate was calculated as a percentage using the formula: (a/b) × 100%, where $a$ represents the number of antibiotics to which an isolate was resistant, and $b$ refers to the total number of antibiotics tested (57, 58). Statistical significance was determined by considering $P$-values with a 95% CI ($P \leq 0.05$). The results are presented in tables and figures, as per the type of data analyzed.

## RESULTS

### Enumeration of coliform bacteria from River Sosiani and wastewater

Total coliforms detected in river water and wastewater from the nine sampling sites were too numerous to count (Table 1). Fecal coliform bacteria levels in river water and wastewater were particularly high at MTRH, Huruma quarry influent, Kipkenyo Boundary influent, and Eldoret Prison, with colony-forming units (CFU) exceeding those observed at Kipkaren Bridge (235), Pioneer Bridge (1,000), Huruma quarry effluent (1,400), and Kipkenyo Boundary effluent, which had up to 10,000 fecal coliforms (Table 1).

**TABLE 1** General analysis of coliform bacteria from River Sosiani and wastewater environments[a]

| S/no | Sample<br>*n* = 9 | Ref. no/ sample code | Water source | Results<br>**Total coliforms fecal coliforms**<br>**(CFU/100 mL)** | |
|---|---|---|---|---|---|
| 1 | Moi Teaching and Referral Hospital (MTRH) | 1384-23/24 | Treated wastewater | TNTC/1 mL | TNTC/100 mL |
| 2 | Huruma quarry influent | 1385-23/24 | Untreated sewage water | TNTC/1 mL | TNTC/100 mL |
| 3 | Huruma quarry effluent | 1387-23/24 | Treated wastewater | TNTC/1 mL | 1,400/100 mL |
| 4 | Pioneer bridge | 1389-23/24 | River | TNTC/1 mL | 1,000/100 mL |
| 5 | Kipkaren bridge | 1390-23/24 | River | TNTC/1 mL | 235/100 mL |
| 6 | Kipkenyo boundary effluent | 1391-23/24 | Treated wastewater | TNTC/1 mL | 10,000/100 mL |
| 7 | Kipkenyo boundary influent | 1392-23/24 | Untreated wastewater | TNTC/1 mL | TNTC/100 mL |
| 8 | Outspan-Nairobi bridge | 1393-23/24 | River | TNTC/1 mL | TNTC/100 mL |
| 9 | Eldoret prison | 1412-23/24 | Wastewater | TNTC/1 mL | TNTC/100 mL |

[a]S/No-serial number; TNTC, too numerous to count; CFU, colony-forming unit. Fecal coliforms are bacterial species found in fecal-polluted water systems, whereas total coliforms are bacterial species found from all kinds of water-polluted systems.

## Identification of bacterial isolates

Gram-negative pathogenic bacteria were isolated using standard culture methods (Fig. 2). The bacterial species identified included *E. coli, Enterobacter aerogenes, Citrobacter spp., Klebsiella spp., Proteus spp., Pseudomonas spp., Salmonella spp.,* and *Yersinia spp.* (Fig. 2). All 10 gram-negative bacteria species, including *E. coli, E. aerogenes, Citrobacter*

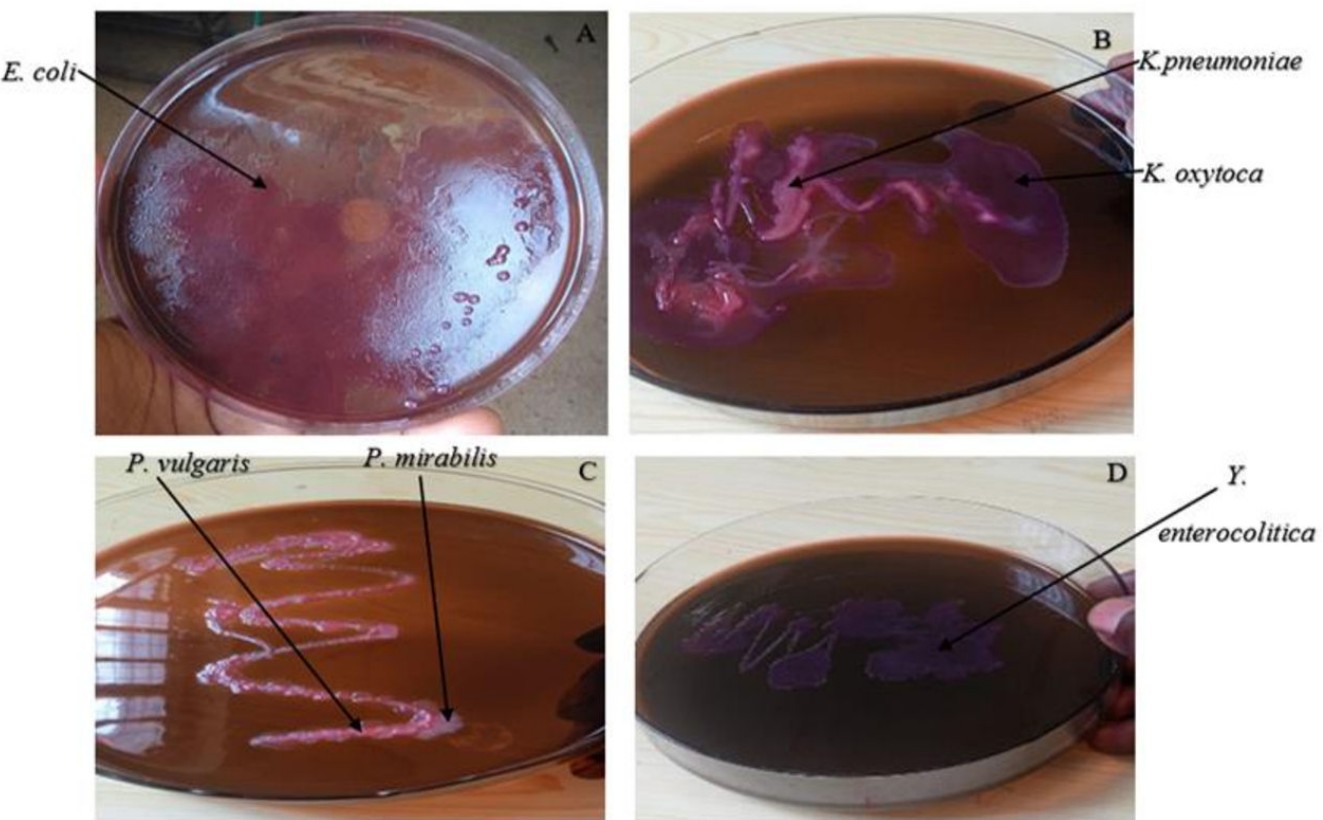

**FIG 2** Bacteria isolation and identification from River Sosiani and wastewater environments on Eosin methylene blue (EMB) agar plates. *E. coli* was characterized as deep purple colonies with a green metallic sheen (A). *K. pneumoniae* appeared as mucoid pink colonies, and *K. oxytoca* appeared as mucoid purple colonies. (B) *P. vulgaris* appeared as pinkish colonies. (C) *P. mirabilis* appeared as translucent colonies, and *Y. enterocolitica* appeared as dark purple colonies without green metallic sheen (D).

*freundii, Klebsiella pneumoniae, Klebsiella oxytoca, Proteus vulgaris, Proteus mirabilis, P. aeruginosa, Salmonella enteritidis,* and *Yersinia enterocolitica,* were rod-shaped (Fig. 3).

Subspecies identification was carried out through five biochemical tests (Fig. 4). The catalase test was used to identify *C. freundii* and *Y. enterocolitica* (Table 2). The citrate test confirmed the presence of *K. oxytoca*, *P. mirabilis*, and *S. enteritidis* (Table 2). The oxidase test was used to detect *P. aeruginosa* (Table 2), whereas the indole test confirmed *P. vulgaris* and *Y. enterocolitica* (Table 2). The triple sugar iron (TSI) test was used to confirm the presence of *K. pneumoniae* (Table 2).

## General analysis of bacteria isolates from the nine sites

The total count of all the bacteria isolates from the nine sites was 1,441, with a *P* value of 0.06 (Table 3). Kipkaren bridge was leading with 296 bacteria isolates, followed by Huruma quarry influent (284), Outspan-Nairobi bridge (218), MTRH (184), Pioneer bridge (172), Eldoret prison (139), Huruma quarry effluent (101), Kipkenyo boundary influent (26), and, finally, Kipkenyo boundary effluent (21) (Table 3).

## Distribution of gram-negative pathogenic bacterial species

Kipkaren Bridge had a higher abundance of bacteria species, comprising up to seven species (Fig. 5). These were *E. coli* (27%), *E. aerogenes* (26%), *K. pneumoniae* (15.9%), *C. freundii* (13.2%), *P. mirabilis* (7.4%), *K. oxytoca* (6.4%), and *P. vulgaris* (4.1%) (Fig. 5). Second, the Huruma quarry influent site had six bacteria species, with *E. coli* occurring at an abundance of 25.7%, followed by *E. aerogenes* (23.2%), *Y. enterocolitica* (16.9%), *S. enteritis* (13.4%), *P. aeruginosa* (12.3%), and *P. mirabilis* (8.6%) (Fig. 5). The Outspan-Nairobi bridge had six species, with *E. coli* dominating the site with an abundance of 39%, followed

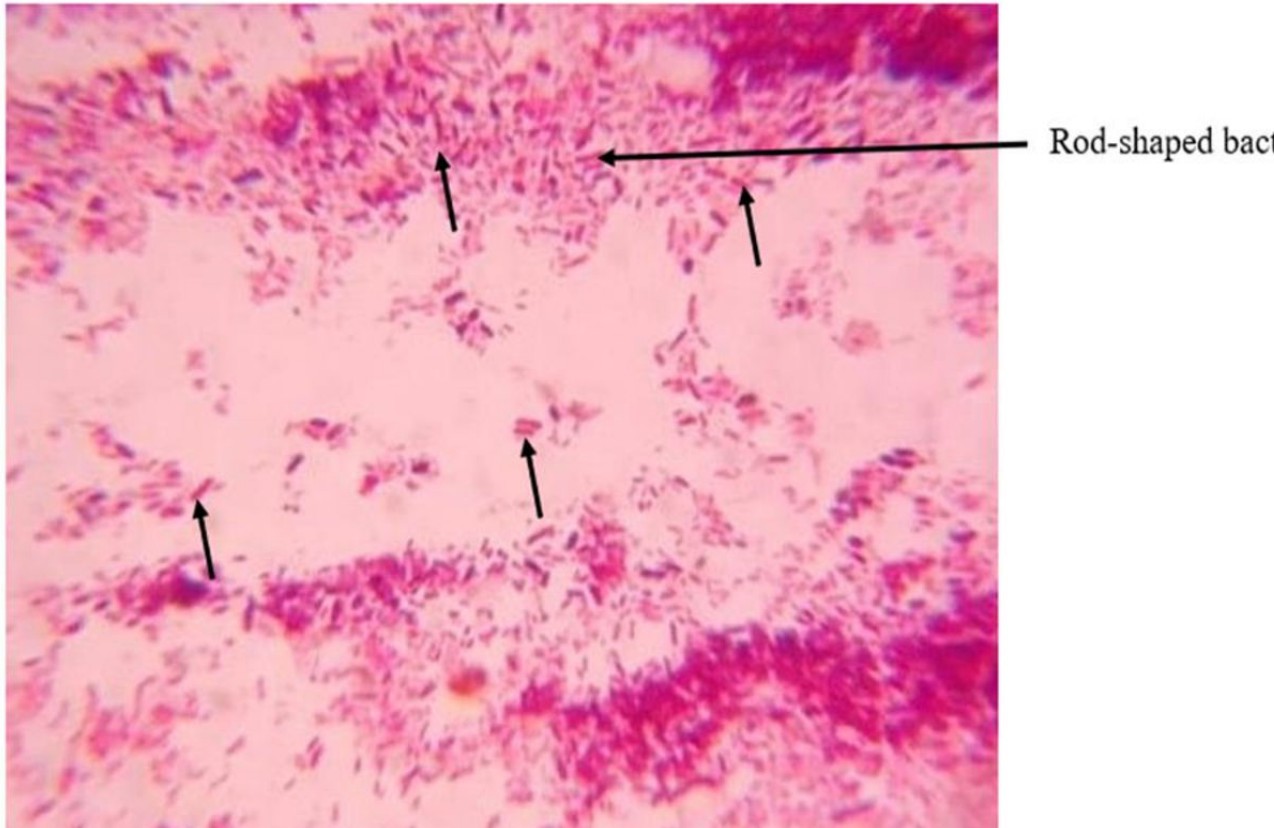

**FIG 3** Microscopic view of the rod-shaped bacterial isolates identified on Gram staining. Isolates of *E. coli, E. aerogenes, C. freundii, K. pneumoniae, K. oxytoca, P. vulgaris, P. mirabilis, P. aeruginosa, S. enteritidis,* and *Y. enterocolitica* were rod-shaped.

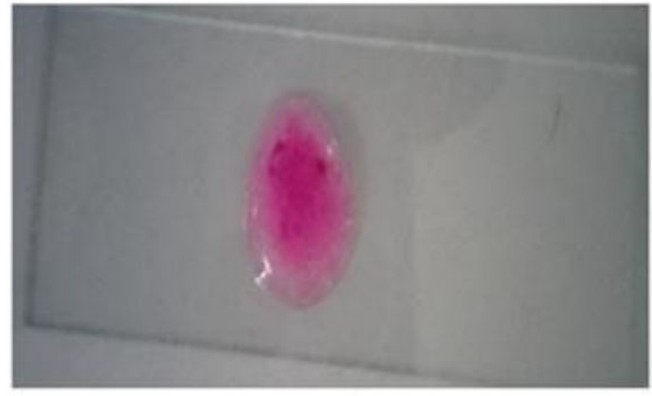

Catalase test

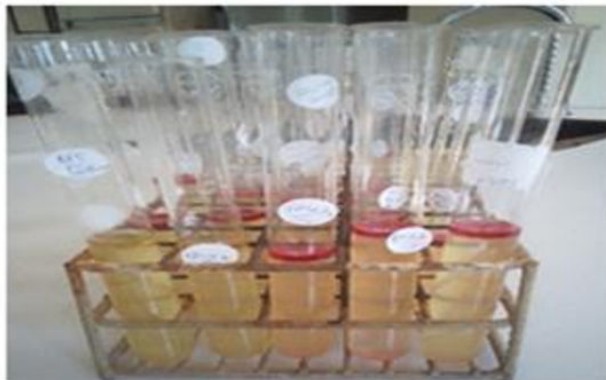

Indole test

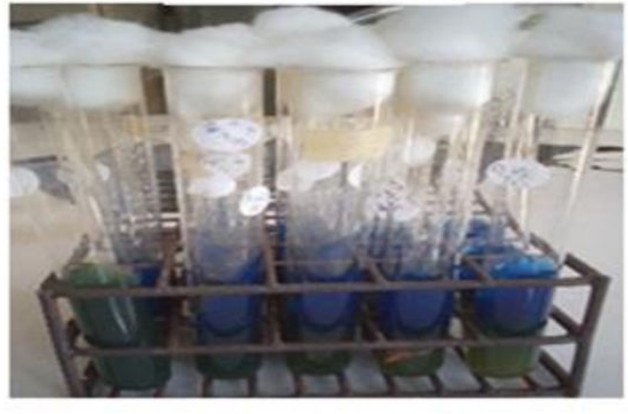

Citrate test

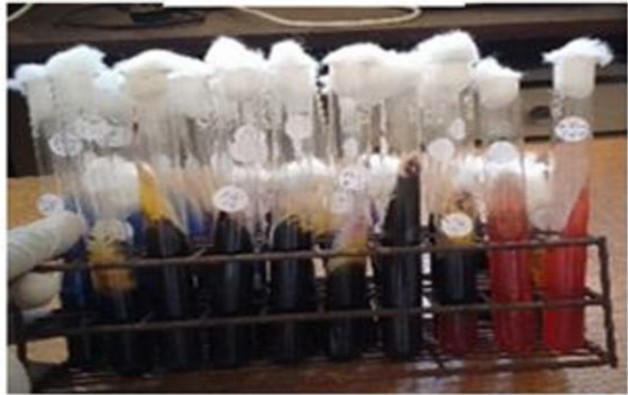

Triple sugar iron test

**FIG 4** Different biochemical tests used in bacteria identification. Bacteria showed a positive catalase test via bubble formation, and a positive indole test was indicated by a red ring on the surface. Bacteria showed a positive citrate test by indicating a blue slant on top of the green butt, and for a triple sugar iron test, a positive reaction was indicated by a yellow/blue slant with a black butt by producing hydrogen sulfide gas and acid.

by *K. pneumoniae* (24.3%), *C. freundii* (19.7%), *P. vulgaris* (7.3%), *E. aerogenes* (6%), and *K. oxytoca* (3.7%) (Fig. 5). Eldoret prison site was detected with five bacteria species, with *S. enteritis* in abundance of 30.2%, followed by *Y. enterocolitica* (23%), *P. aeruginosa* (20.9%), *E. coli* (19.4%), and *P. vulgaris* (6.5%) (Fig. 5). Moreover, the Pioneer bridge site

**TABLE 2** Bacterial isolates from Gram staining and biochemical tests[a]

| S/no | Biochemical test | Sample test characteristic | Bacteria identified | Inference |
|------|------------------|----------------------------|---------------------|-----------|
|      | *n* = 5          |                            |                     |           |
| 1    | Catalase         | Bubble formation           | *C. freundii* and *Y. enterocolitica* | (45) |
|      |                  |                            | *n* = 2             |           |
| 2    | Citrate          | Blue slant on top of green butt | *K. oxytoca. P. mirabilis* and *S. enteritidis* | (46) |
|      |                  |                            | *n* = 3             |           |
| 3    | Oxidase          | Purple colonies            | *P. aeruginosa*     | (47)      |
|      |                  |                            | *n* = 1             |           |
| 4    | Indole           | Reddish ring on the surface | *P. vulgaris* and *Y. enterocolitica* | (48) |
|      |                  |                            | *n* = 2             |           |
| 5    | Triple sugar iron | Yellow/blue slant with black butt | *K. pneumoniae* | (44) |
|      |                  |                            | *n* = 1             |           |

[a]S/No- serial number, *n* = number of biochemical tests/bacterial species identified.

**TABLE 3** Analysis of variation in the number of bacterial isolates detected from the nine sampling sites

| Analysis of variance: single factor | | | | | | |
|---|---|---|---|---|---|---|
| **Groups** | **Count** | **Sum** | **Average** | **Variance** | | |
| Moi Teaching and Referral Hospital | 10 | 184 | 18.4 | 956.4888889 | | |
| Huruma quarry influent | 10 | 284 | 28.4 | 796.4888889 | | |
| Huruma quarry effluent | 10 | 101 | 10.1 | 273.8777778 | | |
| Pioneer bridge | 10 | 172 | 17.2 | 347.9555556 | | |
| Kipkaren bridge | 10 | 296 | 29.6 | 920.7111111 | | |
| Outspan-Nairobi bridge | 10 | 218 | 21.8 | 846.6222222 | | |
| Eldoret Prison | 10 | 139 | 13.9 | 278.5444444 | | |
| Kipkenyo Boundary influent | 10 | 26 | 2.6 | 30.93333333 | | |
| Kipkenyo Boundary effluent | 10 | 21 | 2.1 | 44.1 | | |
| ANOVA | | | | | | |
| Source of Variation | SS | Df | MS | F | *P*-value | F crit |
| Between Groups | 7915.488889 | 8 | 989.4361111 | 1.980755162 | 0.059237593 | 2.054881624 |
| Within Groups | 40461.5 | 81 | 499.5246914 | | | |
| Total | 48376.98889 | 89 | | | | |

reported five species, including *S. enteritis*, *E. aerogenes*, *E. coli*, *C. freundii,* and *P. mirabilis* that were detected at abundances of 26.2%, 20.9%, 18.6%, 18%, and 16.3%, respectively (Fig. 5). In addition, the Moi Teaching and Referral Hospital (MTRH) waste discharge site constituted four species, with *E. coli* being the most abundant at 49.2%, *E. aerogenes* was detected at 34.8%, *P. vulgaris* and *K. oxytoca* had relatively lower abundance values of 12.5% and 6.5%, respectively (Fig. 5). Furthermore, the Huruma quarry effluent had three bacteria, with *E. coli* at an abundance of 39.6%, followed by *E. aerogenes* (33.7%) and *K. pneumoniae* (26.7%) (Fig. 5). The two species of *E. coli* and *E. aerogenes* were found at the Kipkenyo boundary influent with higher abundance values of 57.7% and 42.3%, respectively (Fig. 5). The Kipkenyo boundary effluent was only harbored by *K. pneumoniae* at 100% abundance (Fig. 5).

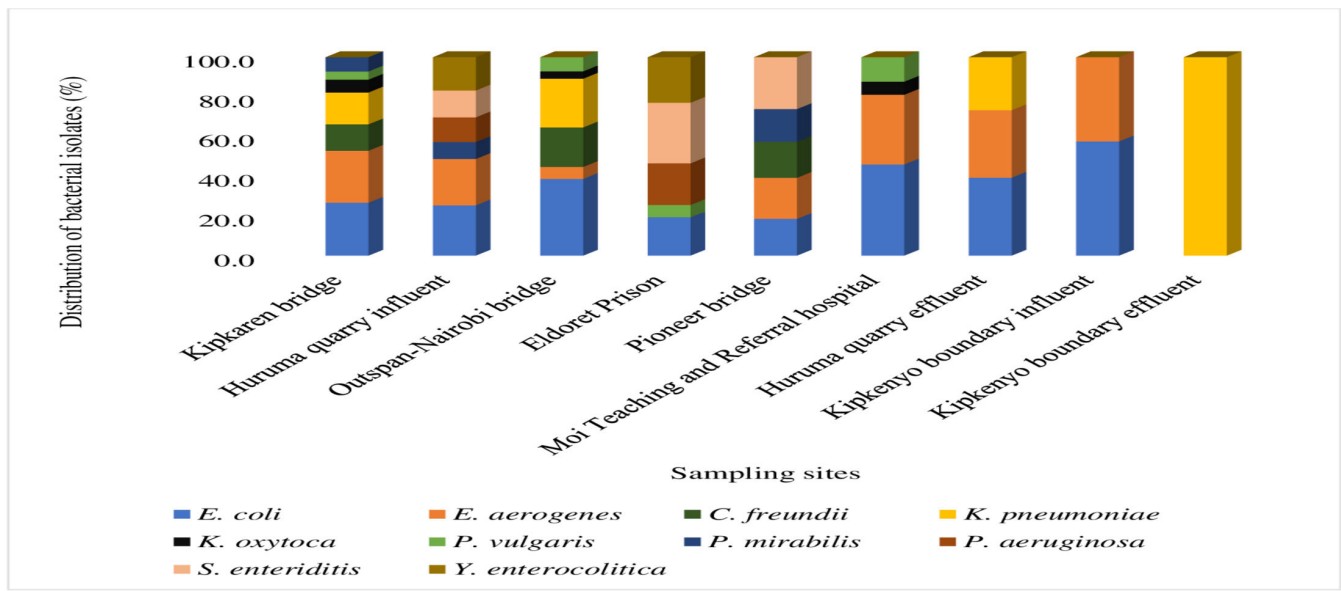

**FIG 5** Distribution of bacterial species in River Sosiani and Wastewater systems. Kipkaren bridge had higher bacterial species count, constituting 7/10 bacteria species, with *E. coli* dominating at a higher abundance of 27% and *P. vulgaris* at a lower abundance of 4.1%. Huruma quarry influent had 6/10 bacteria species, with *E. coli* dominating at a higher abundance of 25.7% and *P. mirabilis* at a lower abundance of 8.6%. Outspan-Nairobi bridge also had 6/10 bacteria species, with *E. coli* dominating at a higher abundance of 39% and *K. oxytoca* at a lower abundance of 3.7%. Both Eldoret prison and Pioneer bridge had 5/10, Moi Teaching and Referral Hospital had 4/10, Huruma quarry effluent had 3/10, Kipkenyo boundary influent constituted 2/10, and finally, Kipkenyo boundary effluent had only *K. pneumoniae* detected at 100% abundance.

In general, *E. coli* and *E. aerogenes* were the most commonly occurring bacteria detected in all the sites at frequencies of 30.3% and 20.9%, respectively (Table 4). *K. pneumoniae* (10.3%), *S enteritidis* (8.7%), *C. freundii* (7.8%), *Y. enterocolitica* (5.6%), *P. mirabilis* (5.1%), *P. aeruginosa* (4.4%), *P. vulgaris* (4.2%), and *K. oxytoca* (2.7%) were also detected at different frequencies (Table 4).

## Antibiotic susceptibility of bacterial isolates to the selected antibiotics

The gram-negative bacterial species had varying susceptibility and resistance profiles (Fig. 6). *E. coli* was susceptible to four antibiotics, doxycycline, ciprofloxacin, gentamicin, and chloramphenicol, with larger zones of inhibition measured at $29 \pm 2.4$ mm, $22.8 \pm 4.1$ mm, $18.6 \pm 3.7$ mm, and $17.4 \pm 3.6$ mm, respectively (Fig. 7). The isolates exhibited intermediate susceptibility to azithromycin ($9.7 \pm 6.8$ mm), minocycline ($8.6 \pm 7.5$ mm), tetracycline ($7 \pm 6.1$ mm), and sulfamethoxazole-trimethoprim ($5 \pm 6.3$ mm), indicating smaller zones of inhibition (Fig. 7). Notably, *E. coli* showed 100% resistance to 12 antibiotics, including penicillin, ampicillin, oxacillin, ceftazidime, erythromycin, co-trimoxazole, amoxicillin-clavulanic acid, streptomycin, kanamycin, levofloxacin, metronidazole, and vancomycin, as no inhibition was observed for these antibiotics (Fig. 7). *E. aerogenes* isolates displayed susceptibility to ciprofloxacin ($25 \pm 3.6$ mm), doxycycline ($23.3 \pm 3.4$ mm), gentamicin ($14.3 \pm 3.3$ mm), chloramphenicol ($12.3 \pm 2.1$ mm), and azithromycin ($10.3 \pm 1.2$ mm) with larger zones of inhibition and intermediate susceptibility to streptomycin ($7.3 \pm 5.3$ mm), tetracycline ($6 \pm 5.7$ mm), and ceftazidime ($2.7 \pm 3.7$ mm) with smaller zones of inhibition (Fig. 8). They were found to exhibit resistance to ampicillin, amoxicillin-clavulanic acid, and sulfamethoxazole-trimethoprim (Fig. 8). *C. freundii* isolates were susceptible to ciprofloxacin, doxycycline, gentamicin, and ceftazidime with larger inhibition zones ($29.3 \pm 1.1$ mm, $25.3 \pm 2.5$ mm, $20 \pm 2$ mm, and $14.3 \pm 4$ mm), respectively (Fig. 9). *C. freundii* isolates were also intermediate susceptible to streptomycin ($9.7 \pm 3.1$ mm), erythromycin ($7.3 \pm 6.4$ mm), and cotrimoxazole ($6.7 \pm 6.1$ mm) with smaller zones of inhibition (Fig. 9). However, *C. freundii* showed 100% resistance to penicillin, ampicillin, and amoxicillin-clavulanic acid (Fig. 9). The isolates of *K. pneumoniae* were susceptible to ciprofloxacin ($17.7 \pm 2.5$ mm), gentamicin ($15.3 \pm 6.4$ mm), azithromycin ($15 \pm 3$ mm), doxycycline ($11.7 \pm 3.5$ mm), and chloramphenicol ($10.3 \pm 9$ mm) with larger zones of inhibition and intermediate susceptible to streptomycin ($6.7 \pm 6.1$ mm), tetracycline ($4 \pm 6.9$ mm), and sulfamethoxazole-trimethoprim ($3.3 \pm 5.8$ mm) with smaller zones of inhibition (Fig. 10). Moreover, the isolates were resistant to ampicillin, amoxicillin-clavulanic, and ceftazidime (Fig. 10).

*Klebsiella oxytoca* isolates exhibited susceptibility to ciprofloxacin, gentamicin, and chloramphenicol, with larger inhibition zones of $29.3 \pm 2.5$ mm, $26.7 \pm 3.2$ mm, and $22.3 \pm 2.5$ mm, respectively (Fig. 11). They displayed intermediate susceptibility to streptomycin ($10 \pm 9.2$ mm), cotrimoxazole ($10 \pm 8.7$ mm), tetracyclines ($5.3 \pm 9.2$ mm), and

**TABLE 4** An overview of bacteria species detected from River Sosiani and wastewater systems at different frequencies[a]

| S/no | Bacteria species | Frequency (%) |
| --- | --- | --- |
|  | *n* = 10 |  |
| 1. | *E. coli* | 30.3 |
| 2. | *E. aerogenes* | 20.9 |
| 3. | *K. pneumoniae* | 10.3 |
| 4. | *S. enteritidis* | 8.7 |
| 5. | *C. freundii* | 7.8 |
| 6. | *Y. enterocolitica* | 5.6 |
| 7. | *P. mirabilis* | 5.1 |
| 8. | *P. aeruginosa* | 4.4 |
| 9. | *P. vulgaris* | 4.2 |
| 10. | *K. oxytoca* | 2.7 |

[a]S/No, serial number, *n* = number of bacteria species detected.

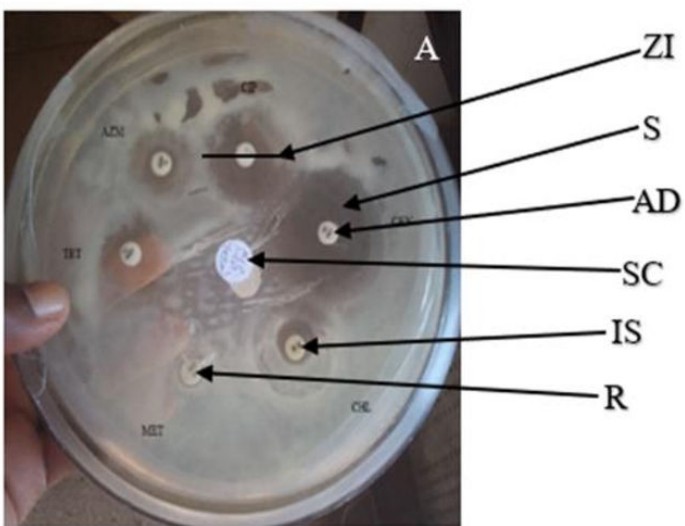
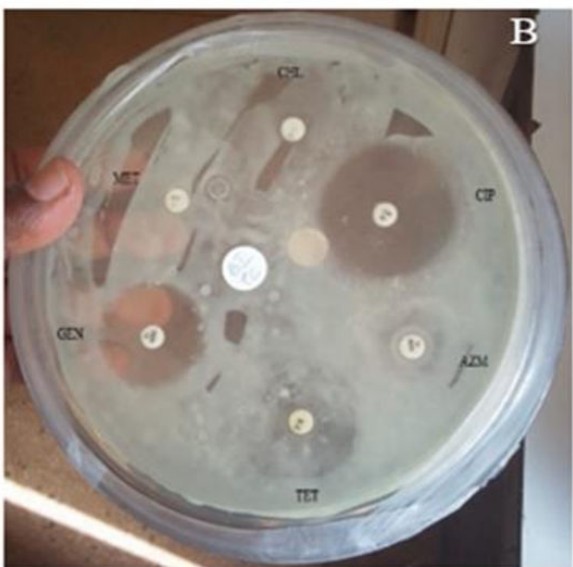

**FIG 6** Antibiotic susceptibility tests of different bacterial isolates on Mueller-Hinton Agar plates. Mueller-Hinton Agar plate (A) consists of *E. coli* isolates subjected to ciprofloxacin, gentamicin, chloramphenicol, metronidazole, tetracycline, and azithromycin. Plate (B) has *Y. enterocolitica* subjected to ciprofloxacin, azithromycin, tetracycline, gentamicin, metronidazole, and chloramphenicol. Plates with complete clear patterns show bacteria that are most susceptible to antibiotics, whereas those with incomplete clear patterns show bacteria that are intermediate susceptible to the tested antibiotics. Plates without any clear patterns show bacteria that are most resistant to the tested antibiotics. ZI, zone of inhibition measured; S, tested antibiotic showed bacterial susceptibility; AD, antibiotic disc; SC, sample code; IS, tested antibiotic showed bacterial intermediate susceptibility, and R, tested antibiotic indicated bacterial resistance.

amoxicillin-clavulanic acid (2.7 ± 4.6 mm) (Fig. 11). Resistance was observed for penicillin, ampicillin, and sulfamethoxazole-trimethoprim (Fig. 11).

*P. vulgaris* isolates were susceptible to ciprofloxacin (28.7 ± 1.2 mm), doxycycline (27.7 ± 2.5 mm), erythromycin (15.7 ± 4 mm), azithromycin (13.7 ± 2.1 mm), and chloramphenicol (11.3 ± 4.2 mm) with larger zones of inhibition (Fig. 12). The isolates demonstrated intermediate susceptibility to cotrimoxazole (7.3 ± 6.4 mm), ceftazidime (6 ± 5.3 mm), tetracycline (4 ± 6.9 mm), amoxicillin-clavulanic acid (3.3 ± 5.8 mm), and gentamicin (3.3

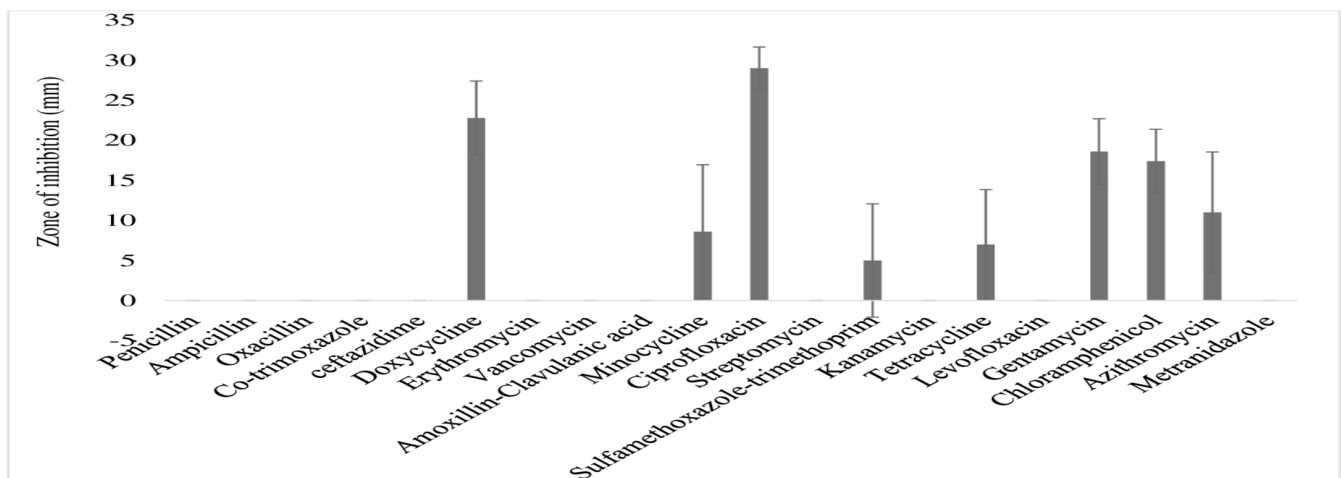

**FIG 7** Antibiotic susceptibility, intermediate susceptibility, and resistance in *E. coli*. The bars represent average values of zones of inhibition per Mueller-Hinton Agar plate, whereas error bars represent the standard deviation obtained from the measured zones of inhibition. Antibiotics with longer bars indicate larger zones of inhibition/complete bacteria clearance. Antibiotics with shorter bars indicate smaller zones of inhibition/incomplete bacterial clearance, whereas those antibiotics with no bars indicate bacteria resistance, that is, penicillin, ampicillin, oxacillin, ceftazidime, erythromycin, cotrimoxazole, amoxicillin-clavulanic acid, streptomycin, kanamycin, levofloxacin, metronidazole, and vancomycin, could not inhibit *E. coli* growth and are thus considered ineffective.

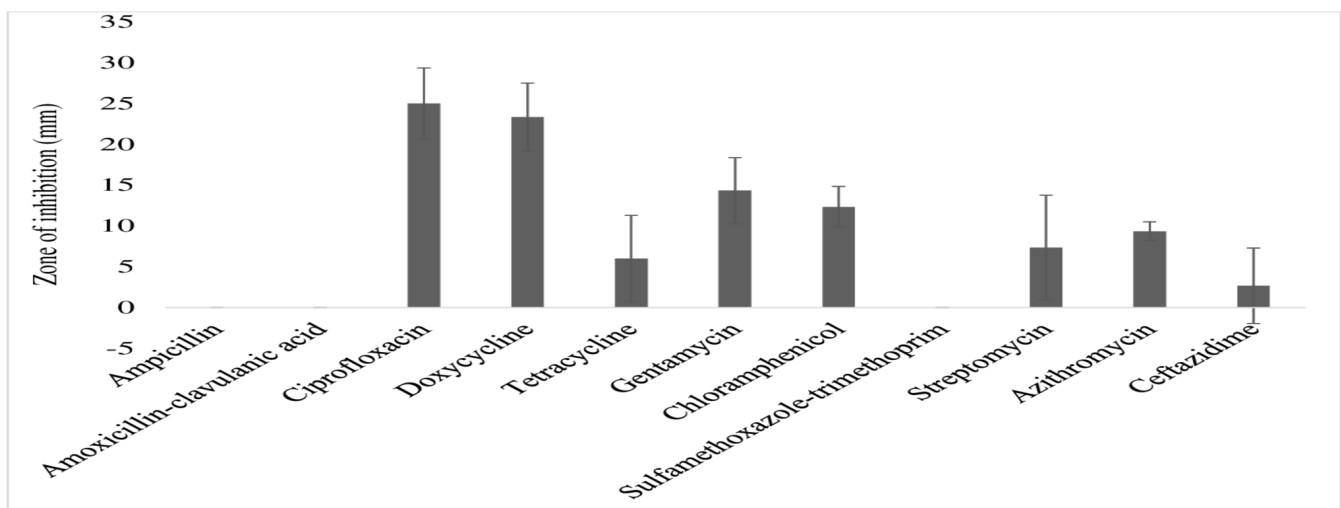

**FIG 8** Antibiotic susceptibility, intermediate susceptibility, and resistance in *E. aerogenes*. The bars represent average values of zones of inhibition per Mueller-Hinton Agar plate, whereas error bars represent the standard deviation obtained from the measured zone of inhibition. Antibiotics with longer bars indicate larger zones of inhibition/complete bacteria clearance. Antibiotics with shorter bars indicate smaller zones of inhibition/incomplete bacteria clearance, whereas those antibiotics with no bars indicate bacteria resistance, that is, ampicillin, amoxicillin-clavulanic acid, and sulfamethoxazole-trimethoprim could not inhibit *E. aerogenes* growth and are thus considered ineffective.

± 5.8 mm) with smaller zones of inhibition (Fig. 12). *P. vulgaris* was resistant to ampicillin and streptomycin (Fig. 12). *P. mirabilis* demonstrated susceptibility to ciprofloxacin (22.7 ± 2.1 mm), doxycycline (22 ± 2 mm), gentamicin (20 ± 2 mm), and chloramphenicol (11 ± 9.6 mm) with larger zones of inhibition (Fig. 13). They were intermediate susceptible to streptomycin (9.3 ± 8.1 mm), azithromycin (6 ± 10.4 mm), tetracycline (4 ± 6.9 mm), erythromycin (3.3 ± 5.1 mm), and amoxicillin-clavulanic acid (2.7 ± 4.6 mm) with smaller zones of inhibition (Fig. 13). *P. mirabilis* was resistant to ampicillin, ceftazidime, and cotrimoxazole, showing no inhibition for these antibiotics (Fig. 13).

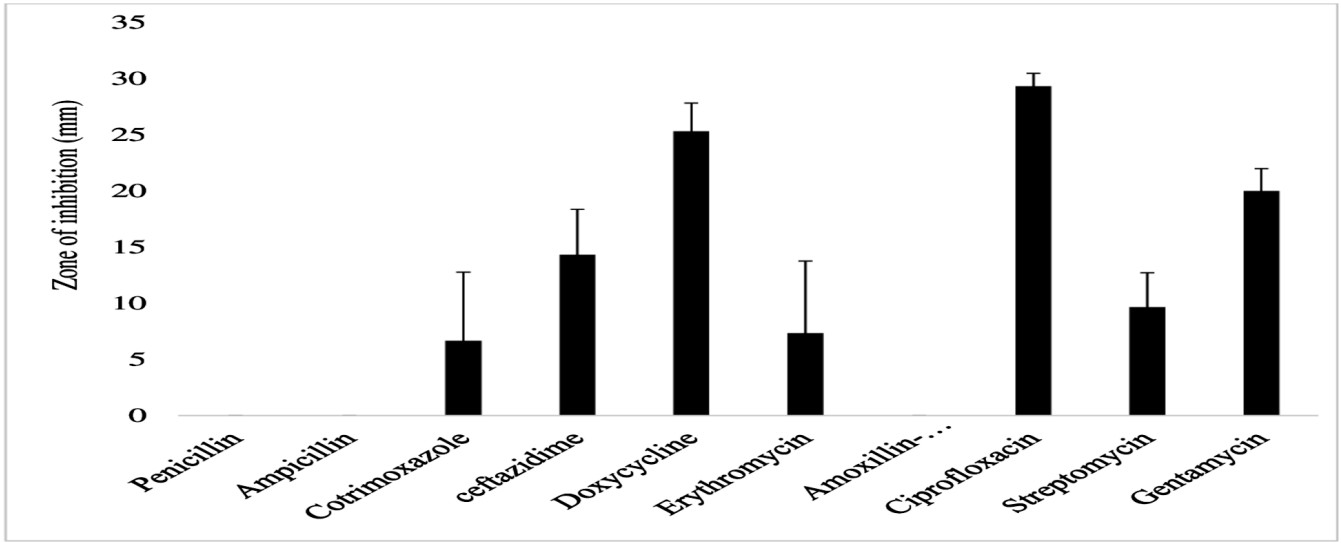

**FIG 9** Antibiotic susceptibility, intermediate susceptibility, and resistance in *C. freundii*. The bars represent the average values of zones of inhibition per Mueller-Hinton Agar plate, whereas the error bars represent the standard deviation obtained from the measured zone of inhibition. Antibiotics with longer bars indicate larger zones of inhibition/complete bacterial clearance. Antibiotics with shorter bars indicate smaller zones of inhibition/incomplete bacterial clearance, whereas those antibiotics with no bars indicate bacteria resistance, that is, penicillin, ampicillin, and amoxicillin-clavulanic acid could not inhibit *C. freundii* growth and are thus considered ineffective.

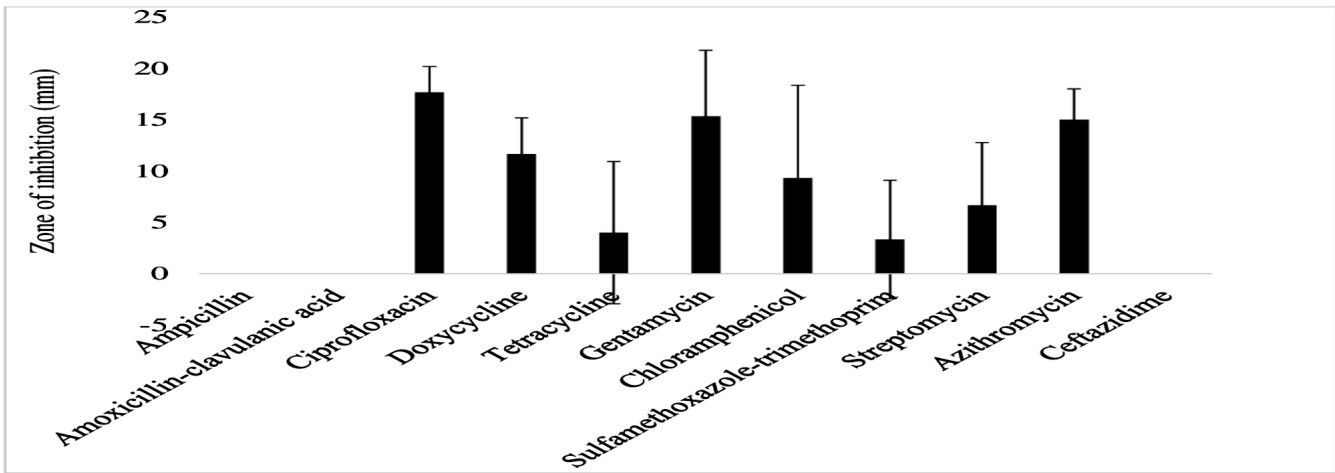

**FIG 10** Antibiotic susceptibility, intermediate susceptibility, and resistance patterns detected in *K. pneumoniae.* The bars represent average values of zones of inhibition per Mueller-Hinton Agar plate, whereas the error bars represent the standard deviation obtained from the measured zone of inhibition. Antibiotics with longer bars indicate a larger zone of inhibition/complete bacterial clearance. Antibiotics with shorter bars indicate smaller zone of inhibition/incomplete bacteria clearance, whereas those antibiotics with no bars indicate bacteria resistance, that is, ampicillin, amoxicillin-clavulanic, and ceftazidime could not inhibit *K. pneumoniae* growth and are thus considered ineffective.

Similarly, *P. aeruginosa* exhibited susceptibility to ciprofloxacin (22.3 ± 5.9 mm) and chloramphenicol (13.3 ± 6.1 mm) with larger zones of inhibition but displayed resistance to sulfamethoxazole-trimethoprim (Fig. 14). The bacteria had smaller inhibition zones for other antibiotics tested, including ceftazidime (7.3 ± 6.4 mm), gentamicin (6.7 ± 5.8 mm), tetracycline (5.3 ± 9.2 mm), azithromycin (3.3 ± 5.8 mm), and streptomycin (2.7 ± 4.6 mm) (Fig. 14). *Salmonella enteritidis* was susceptible to ciprofloxacin (27.7 ± 2.5 mm), gentamicin (22.3 ± 3.2 mm), chloramphenicol (17.7 ± 4.2 mm), and azithromycin (10.7 ± 1.2 mm) with larger zones of inhibition (Fig. 15). Also, they were intermediate susceptible to tetracycline (4 ± 6.9 mm) and metronidazole (3.3 ± 5.8 mm) with smaller zones of inhibition (Fig. 15). However, no resistance to any of the tested antibiotics was observed in *Salmonella enteritidis* (Fig. 15). Finally, *Y. enterocolitica* was susceptible to

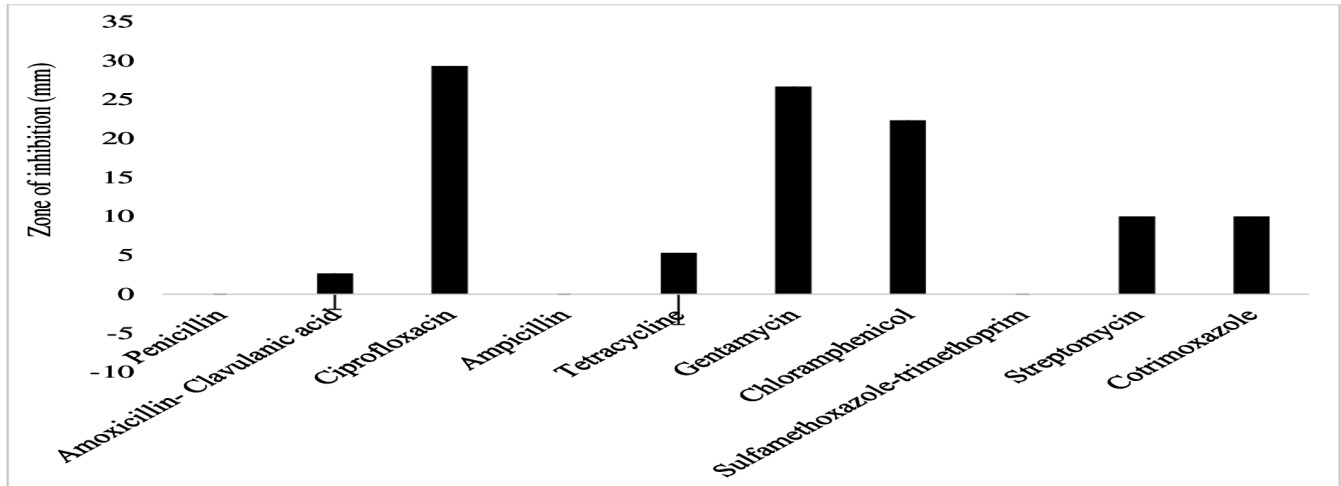

**FIG 11** Antibiotic susceptibility, intermediate susceptibility and resistance in *K. oxytoca.* The bars represent average values of the zones of inhibition per Mueller-Hinton Agar plate, whereas the error bars represent the standard deviation obtained from the measured zones of inhibition. Antibiotics with longer bars indicate larger zones of inhibition/complete bacterial clearance. Antibiotics with shorter bars indicate smaller zones of inhibition/incomplete bacterial clearance, whereas those antibiotics with no bars indicate bacterial resistance, that is, penicillin, ampicillin, and sulfamethoxazole-trimethoprim could not inhibit *K. oxytoca* growth and are thus considered ineffective.

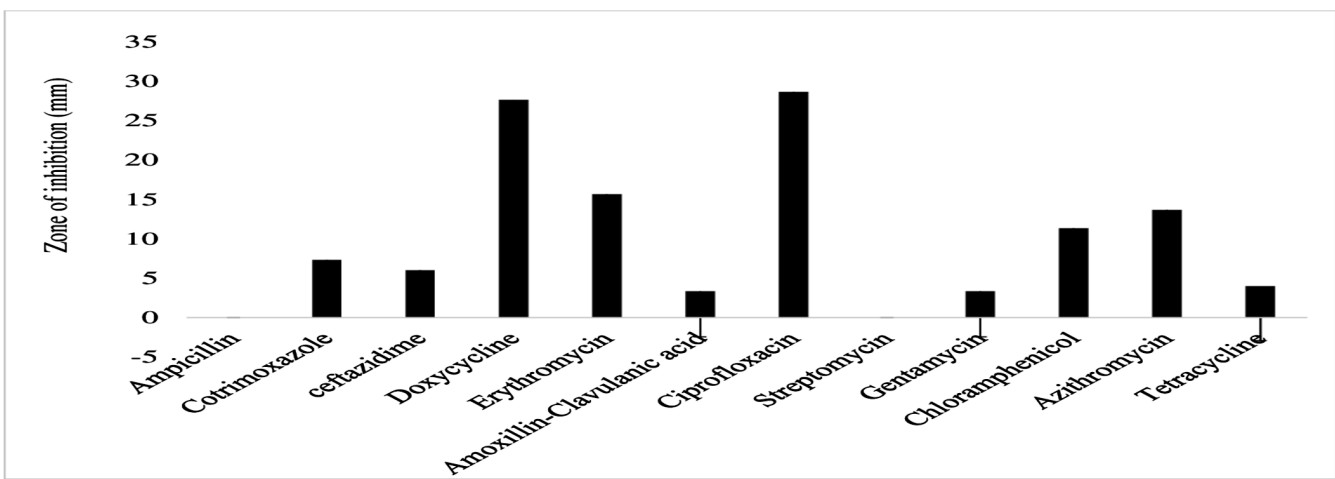

**FIG 12** Antibiotic susceptibility, intermediate susceptibility, and resistance in *P. vulgaris*. The bars represent average values of zones of inhibition per Mueller-Hinton Agar plate, whereas error bars represent the standard deviation obtained from the measured zone of inhibition. Antibiotics with longer bars indicate larger zones of inhibition/complete bacterial clearance. Antibiotics with shorter bars indicate smaller zones of inhibition/incomplete bacterial clearance, whereas those antibiotics with no bars indicate bacterial resistance, that is, ampicillin and streptomycin could not inhibit *P. vulgaris* growth and are thus considered ineffective.

ciprofloxacin (26 ± 1.7 mm), chloramphenicol (22 ± 1 mm), gentamicin (19.3 ± 1.2 mm), tetracycline (15.3 ± 3 mm), and azithromycin (14.7 ± 2.3 mm) with larger zones of inhibition (Fig. 16). The species showed intermediate susceptibility to metronidazole (10 ± 5.2 mm) and amoxicillin-clavulanic acid (6.7 ± 6.1 mm) with smaller zones of inhibition (Fig. 16). Moreover, *Y. enterocolitica* was resistant to ampicillin, showing no inhibition to this antibiotic (Fig. 16).

## Multi antibiotic resistance index in the bacterial isolates

The findings of this study highlight the varying degrees of multiple antibiotic resistance indices (MARI) observed among the isolated bacteria (Fig. 17). *E. coli* exhibited the highest MARI at 60%, showing resistance to 12 antibiotics (Fig. 17). *Enterobacter*

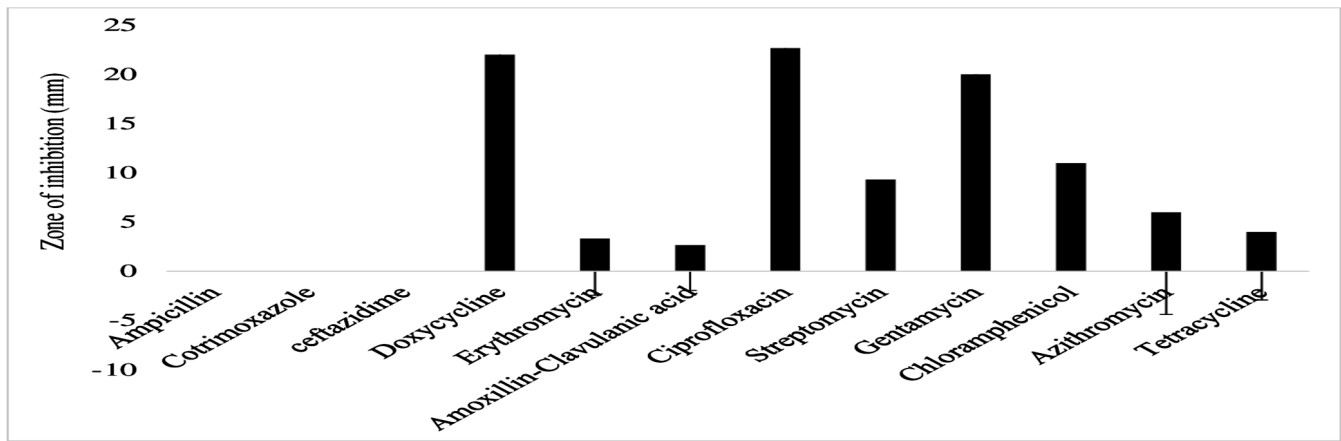

**FIG 13** Antibiotic susceptibility, intermediate susceptibility, and resistance patterns in *P. mirabilis*. The bars represent average values of zones of inhibition per Mueller-Hinton Agar plate, whereas the error bars represent the standard deviation obtained from the measured zone of inhibition. Antibiotics with longer bars indicate larger zones of inhibition/complete bacterial clearance. Antibiotics with shorter bars indicate smaller zones of inhibition/incomplete bacterial clearance, whereas those antibiotics with no bars indicate bacterial resistance, that is, ampicillin, ceftazidime, and cotrimoxazole could not inhibit *P. mirabilis* growth and are thus considered ineffective.

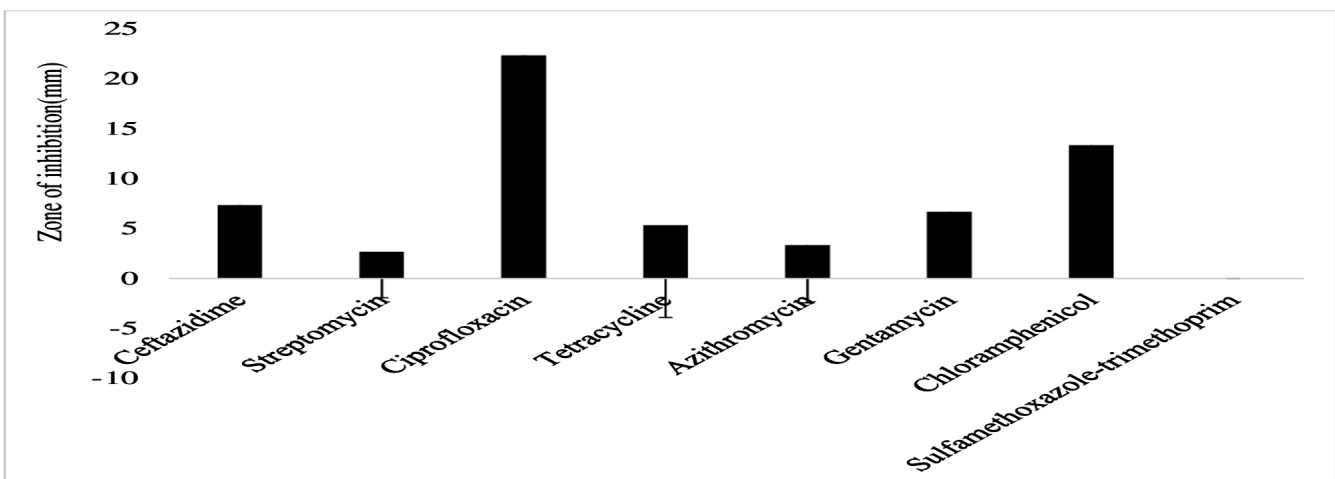

**FIG 14** Antibiotic susceptibility, intermediate susceptibility, and resistance in *P. aeruginosa*. The bars represent the average values of the zones of inhibition per Mueller-Hinton Agar plate, whereas the error bars represent the standard deviation obtained from the measured zone of inhibition. Antibiotics with longer bars indicates larger zones of inhibition/complete bacterial clearance. Antibiotics with shorter bars indicate smaller zones of inhibition/incomplete bacterial clearance, whereas those antibiotics with no bars indicate bacteria resistance, that is, Sulfamethoxazole-trimethoprim could not inhibit *P. aeruginosa* growth and is thus considered ineffective.

*aerogenes* demonstrated a MARI of 33.3%, being resistant to ampicillin, amoxicillin-clavulanic acid, and sulfamethoxazole-trimethoprim (Fig. 17).

*C. freundii* and *K. oxytoca* each exhibited a MARI of 30%, showing resistance to penicillin, ampicillin, amoxicillin-clavulanic acid, and sulfamethoxazole-trimethoprim (Fig. 17). Similarly, *K. pneumoniae* and *P. mirabilis* had a MARI of 25.0%, with resistance to ampicillin, amoxicillin-clavulanic acid, ceftazidime, and co-trimoxazole (Fig. 17). On the other hand, *P. vulgaris*, *P. aeruginosa*, and *Y. enterocolitica* exhibited lower MARI values of 16.7%, 12.5%, and 12.5%, respectively (Fig. 17). These species were resistant to ampicillin, streptomycin, and sulfamethoxazole-trimethoprim (Fig. 17). Notably, *S. enteritidis* showed no resistance to the tested antibiotics, resulting in a MARI of 0% (Fig. 17).

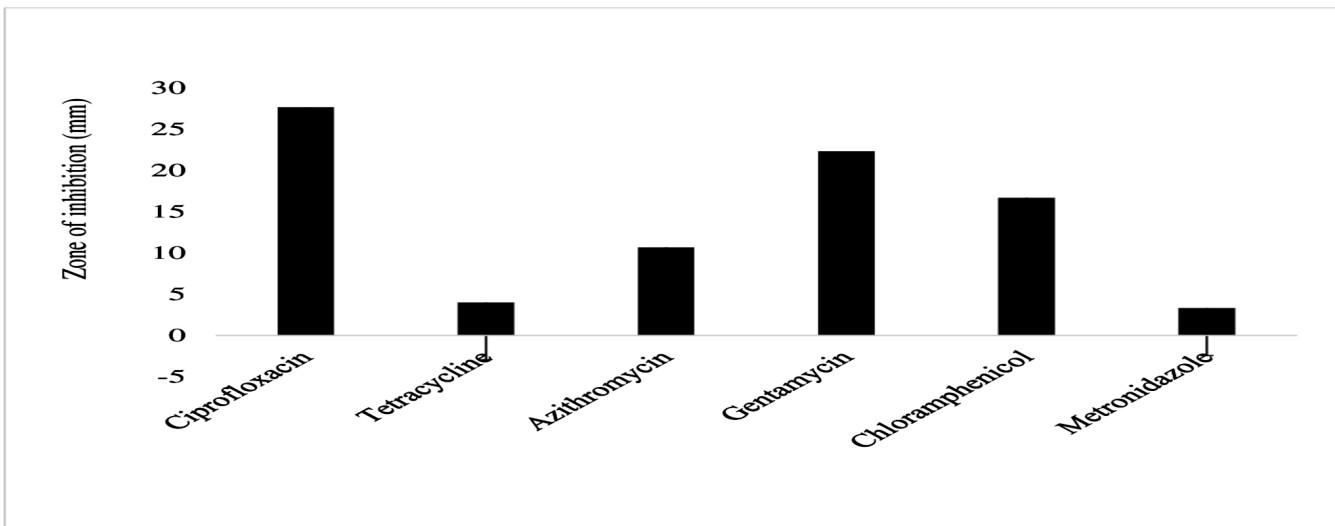

**FIG 15** Antibiotic susceptibility, intermediate susceptibility, and resistance in *S. enteritis*. The bars represent the average values of zones of inhibition per Mueller-Hinton Agar plate, whereas the error bars represent the standard deviation obtained from the measured zones of inhibition. Antibiotics with longer bars indicate larger zones of inhibition/complete bacterial clearance. Antibiotics with shorter bars indicate smaller zones of inhibition/incomplete bacterial clearance. *S. enteritis* isolates showed no resistance to the six tested antibiotics.

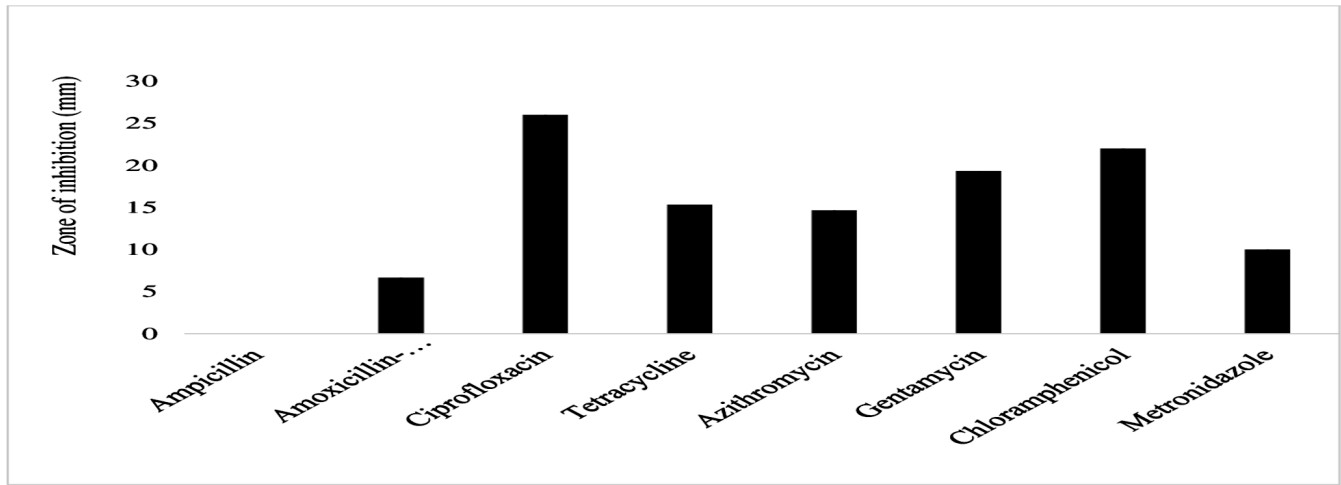

**FIG 16** Antibiotic susceptibility, intermediate susceptibility, and resistance in *Y. enterocolitica*. The bars represent the average values of zones of inhibition per Mueller-Hinton Agar plate, whereas the error bars represent the standard deviation obtained from the measured zones of inhibition. Antibiotics with longer bars indicate larger zones of inhibition/complete bacterial clearance. Antibiotics with shorter bars indicate smaller zones of inhibition/incomplete bacterial clearance, whereas those antibiotics with no bars indicate bacterial resistance, that is, ampicillin could not inhibit *Y. enterocolitica* growth and is thus considered ineffective.

## DISCUSSION

### Enumeration of coliform bacteria from river Sosiani and wastewater

The study reveals that Moi Teaching and Referral Hospital (MTRH), Huruma quarry influent, Kipkenyo boundary influent, Outspan-Nairobi bridge, and Eldoret Prison sites exhibited extremely high bacterial loads, often beyond count (too numerous to count). These locations contained significant amounts of fecal coliforms, suggesting that large quantities of fecal matter were being directly discharged into River Sosiani and surrounding wastewater environments. The presence of fecal coliforms in these sites implies that many residents and occupants in the vicinity lack basic sanitary facilities,

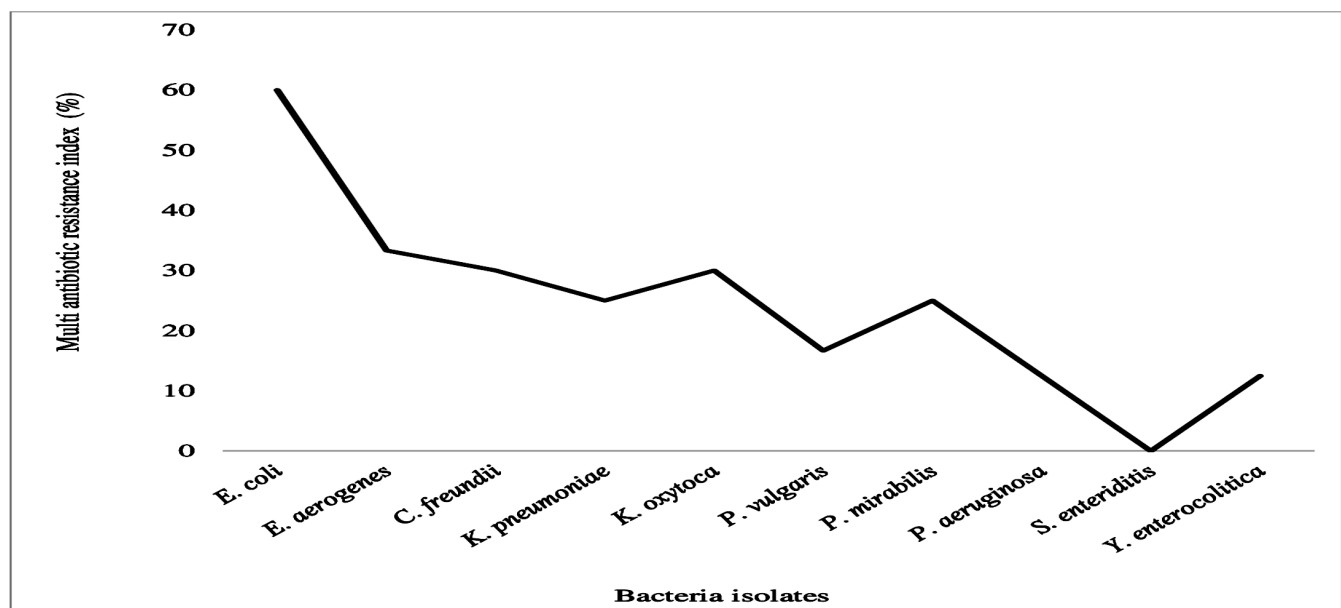

**FIG 17** Multi--antibiotic resistance index of different bacterial isolates. Bacteria with low MARI, including *P. vulgaris, P. aeruginosa,* and *Y. enterocolitica,* were resistant to a few antibiotics. Bacteria with high MARI, including *E. coli, E. aerogenes, C. freundii, K. pneumoniae, K. oxytoca,* and *P. mirabilis,* were resistant to multiple antibiotics. *Salmonella enteritidis* indicated susceptibility to all six tested antibiotics, showing no resistance (0% MARI).

and the wastewater treatment at these facilities is either ineffective or sometimes not performed at all. This situation exposes the local population to serious health risks, particularly those who rely on River Sosiani for drinking water and other domestic uses, as well as animals that drink from the river. Previous studies have similarly documented the dangers posed by fecal coliforms to human, animal, and ecosystem health (59).

Notably, the Kipkenyo boundary effluent site had a relatively higher concentration of fecal coliforms (10,000 CFU/100 mL) compared with the Kipkaren bridge (235 CFU/100 mL), exceeding the World Health Organization's recommended limit of 1,000 CFU/100 mL (60). The effluents from both Pioneer Bridge and Huruma quarry showed fecal coliform counts ranging between 1,000 and 1,400 CFU/100 mL. Huruma quarry and Kipkenyo boundary are both sewage treatment ponds managed by the Eldoret Water and Sanitation Company (ELDOWAS). Kipkenyo effluent displayed a significantly higher fecal coliform load than Huruma quarry, which could indicate more effective sewage treatment at Huruma. This difference may be due to more regular and concentrated chemical applications at Huruma quarry. Additionally, Kipkenyo, being a rural area with fewer resources, likely struggles with less access to sanitary facilities and clean water, contributing to higher fecal contamination. In contrast, the other sites, located within Eldoret town and its surrounding estates, benefit from better economic resources, which typically enable residents to afford better sanitation and clean water, thereby reducing the flow of fecal matter into the environment. Bacterial contamination at the Moi Teaching and Referral Hospital can largely be attributed to the hospital's operations and the nearby commercial center. Medical wastes, such as used bandages, syringes, needles, cotton wool, gloves, and disinfectants, are either incinerated or improperly disposed of, especially liquid wastes, which often find their way into disposal sites and drainage systems without adequate treatment. Although hospital waste undergoes some pre-treatment before being discharged into the municipal sewerage system, which leads to Huruma quarry's sewage treatment plant, the sheer volume of total and fecal coliforms at this site suggests that proper treatment processes may not always be rigorously applied. A similar study (60) found higher concentrations of total coliforms ($40 \times 10^4$ CFU/100 mL) compared with fecal coliforms ($10^4$ CFU/100 mL) in hospital effluent, indicating that hospital wastewater treatment is not always sufficient.

Surface runoff from surrounding agricultural areas and densely populated estates on the outskirts of Eldoret also contributes to bacterial contamination at sites like Outspan-Nairobi Road bridge, Kipkaren Bridge, and Pioneer Bridge, particularly during the rainy season. Rainwater washes waste materials into drainage systems, which eventually flow into rivers like the Sosiani. One study supports these findings, documenting total coliform counts ranging from $10^{7.0-8.0}$ CFU/100 mL from surface runoff (61). Additionally, the activities of nearby establishments, such as guest houses, car washes near Outspan-Nairobi Road bridge, and a petrol station around Pioneer Bridge, likely contribute to bacterial pollution in these areas. Furthermore, Kipkaren Bridge is located near a municipal market, light industries, and motor vehicle service workshops, which, although not directly contributing to bacterial contamination, may still play a role in polluting River Sosiani through runoff and waste discharge.

Bridges themselves can also serve as bacterial reservoirs, receiving contamination from heavy transport vehicles, human traffic, and improperly disposed waste such as carrier tins, plastic bottles, and packaging materials. Domestic users who wash and bath at the bridges further exacerbate contamination levels. This aligns with findings from other studies that have identified bacteria accumulation at sea-crossing and pedestrian bridges in urban areas (62, 63) and from surface runoff into rivers (64). Thus, further research is needed to assess the full extent of bacterial contamination at river bridges and to develop strategies for reducing pathogenic bacterial loads from these sites.

## Isolation and identification of pathogenic bacterial species

In the present study, the culture method proves to be quite effective in the isolation and identification of bacterial groups of different species from wastewater, consistent

with its use in related studies (55). Biochemical tests also proved to be efficient in the identification of specific sub-species of bacteria isolates from water systems. Similar biochemical tests have been used in several studies, as standard practice (46, 65). A total of 1,441 bacteria were isolated with a *P* value of 0.06, indicating that there was no significant difference in the number of bacteria isolates detected across the nine sites. This comprised the following 10 bacterial species: *E. coli, E. aerogenes, C. freundii, K. pneumoniae, K. oxytoca, P. vulgaris, P. mirabilis, P. aeruginosa, S. enteritidis,* and *Y. enterocolitica*. However, gram-positive bacteria also do exist in water environments (38, 66), but the scope of this study was much more on antibiotic resistance in bacteria. Gram-negative bacteria are widely known for spreading bacterial infections to hosts, which are treated using common antibiotics (67). Therefore, it is expected that these pathogenic bacteria also acquire high rates of resistance from the environment toward the antibiotics. Most of these gram-negative isolates detected in this study were similarly reported by other studies to dominate in wastewater systems (68) and other environments (69, 70).

The present investigations demonstrated that *E. coli* and *E. aerogenes* were the most commonly occurring bacterial contaminants detected in all the sites at abundances of 30.3% and 20.9%, respectively. A recent study reports *E. coli* as the most common bacterial isolates, occurring at high numbers in lake water (14). In most of these environments, the two species cause adverse effects, since they are pathogenic. These include severe fecal pollution and antibiotic resistance in the environment (71, 72). Apart from these effects, the two species, in addition to *K. pneumoniae, K. oxytoca, P. vulgaris, P. mirabilis, P. aeruginosa, K. pneumoniae,* and *P. mirabilis* also cause urinary tract infections (UTIs) in humans among other diseases (73–76). Therefore, the presence of *E. coli* and *E. aerogenes* detected in comparatively higher abundances predicts the public health risks to the residents of Uasin Gishu county, especially the people and livestock that directly use water from River Sosiani.

*C. freundii* causes sporadic, nosocomial, neonatal bacterial meningitis, UTIs, severe diarrhea, and brain abscesses in humans (77, 78). It also damages the structural integrity of crayfish muscle in aquatic systems (79), alters the genetic structural diversity of soil microbial community (80), and infects ducks, which later spread more of multidrug-resistant *C. freundii* isolates in the environment (81). Therefore, its presence poses danger to people, livestock, poultry, and even farmlands.

Moreover, a high number of *S. enteritis* and *Y. enterocolitica* were found in MTRH and Eldoret prison. They associate with poultry, slaughterhouses, and livestock products (82–84). Their detection in the current study seems to suggest that these MTRH and Eldoret prison are constantly receiving large amounts of effluents from poultry, slaughterhouses, and other animal husbandry sites. For instance, the MTRH waste disposal screen is surrounded by many residents and located adjacent to the big hotel that serves many people within and outside the hospital. It is possible that the wastes from the slaughtered poultry and meat products get offloaded into the nearby MTRH disposal screen site. Additionally, Eldoret prison appears to be keeping livestock and poultry, whose wastes (meat products and manure), in addition to those from the neighboring Eldoret GK primary and secondary schools, get disposed of into the site. Therefore, contributing to the existence of *S. enteritis* and *Y. enterocolitica* in the environment, which predispose residents to infections of zoonotic diseases, including the veterinaries, meat consumers, slaughter workers, and other subjects handling livestock and poultry sectors in MTRH and Eldoret prison, which may further spread to other parts of the country.

## Distribution of pathogenic bacteria loads in River Sosiani and wastewater environments

Isolates of *E. coli, E. aerogenes,* and *C. freundii* were commonly found in all the three bridges sampled, including Kipkaren bridge (upstream), Pioneer bridge (center), and Outspan-Nairobi bridge (downstream) along River Sosiani. Kipkaren bridge and Outspan-Nairobi bridge sites also had high concentrations of *K. pneumoniae* and *K. oxytoca*. The bacteria species detected at these bridges could be either those circulating

in the river from upstream or the ones boosted by surface runoff, especially during flooding and other sources linked to the usage of the roads and bridges. They also end up introducing other toxic substances in the environment (85–87), which cause AMR in bacteria. Although effects of such activities from roads, bridges, and establishments on loading of wastes to aquatic habitats are least documented, it is possible that they are sources of pathogenic bacteria in rivers and therefore may directly increase the public health risk of using the water from River Sosiani.

This study showed that untreated wastes had more bacteria species than the treated wastes. Similar isolates of *E. coli, E. aerogenes, K. pneumoniae, K. oxytoca,* and *P. vulgaris* were both detected in untreated and treated wastes, indicating that current treatment methods used do not fully eliminate bacteria from wastewater. Nevertheless, there were more pathogenic bacteria species recorded at the influents than effluent sites, suggesting that wastewater treatment plants (WWTP) of Huruma quarry and Kipkenyo boundary often receive more raw wastes from urbanization, domestics, animal husbandry, and agriculture, which contain large bacterial counts. It seems that after the treatment process, the bacterial count reduces, accounting for fewer bacteria observed at the effluent sites. At the influent points, *E. coli, E. aerogenes, P. mirabilis, P. aeruginosa,* and *S. enteritidis* were detected in both WWTPs, whereas *K. pneumoniae* was the only isolate observed dominating the effluents of sewage treatment plants. Globally, the current applied disinfection technologies in most wastewater treatment plants (WWTPs) are chlorination, ultraviolet (UV) radiation, and ozonation (88). Both Huruma quarry and Kipkenyo boundary wastewater treatment plants use naturally occurring aerobic bacteria to remove the biological contaminants and disinfect the remaining influxes with chlorine before discharging clean wastewater into River Sosiani. For instance, Huruma quarry influent had five bacterial species, but after treatment process, only three (*E. coli, E. aerogenes, and K. pneumoniae*) were detected at its effluent point. This shows that although action by aerobic bacteria coupled with chlorine destroys some pathogenic bacteria species (*P. mirabilis, P. aeruginosa, and S. enteritis*), it is not effective against some bacteria (*E. coli* and *E. aerogenes*). This may also suggest these bacterial species are resistant to the natural methods of wastewater treatment used and may therefore require the use of stronger antibiotics or more innovative natural remedies that are not only stronger against bacteria but also environmentally friendly. A recent investigation (88) also reported that *E. coli,* found at the influent and effluent sites of the two different WWTPs, to be resistant to chlorination, which complies with this work. Furthermore, the study suggests that chlorination only is an inefficient disinfection method in WWTPs; hence, it should be coupled with ultraviolet (UV) radiation and ozonation (89), which penetrates to disrupt bacterial cells and degrades DNA, ensuring complete death and elimination of the bacteria. *K. pneumoniae* seems to have colonized on a site far from the influent point and underwent crosstransition within the water toward the effluent sites. Similarly, the biological treatment method at Kipkenyo boundary suggests being more effective than the one used in Huruma quarry sewage treatment plants, since both *E. coli* and *E. aerogenes* at the influent were absent at the effluent site at the Kipkenyo boundary site.

The treated medical wastes from MTRH and untreated raw wastes from Eldoret prison had *E. coli* and *P. vulgaris*. Worldwide, steam sterilization via autoclaving, incineration, plasma gasification, and microwave methods, among others, is recommended and often used in the treatment of medical wastes (90, 91). MTRH often uses the incineration method (92), which is the most common treatment method of medical wastes used worldwide (93). Incineration seems to be a powerful technique operated at extremely high temperatures, hence effective in killing the pathogenic bacteria in wastes. Therefore, the presence of *E. coli, E. aerogenes, K. oxytoca,* and *P. vulgaris* in MTRH waste screen sites is suggested to be generated from the neighboring hotels, car wash, and other commercial enterprises, hence increasing pathogenic bacteria at the site. On the other hand, detection of *S. enteritidis* and *Y. enterocolitica* in untreated wastewater from Eldoret GK prison could be attributed to the lack of proper waste treatments at

the establishment. Despite the general improvement of the conditions and facilities in Kenyans' prison since 2015, congestion due to large number of inmates is still common and seems to contribute significantly to the presence of pathogenic bacteria (*E. coli*, *P. vulgaris*, *P. aeruginosa*, *S. enteritidis,* and *Y. enterocolitica*) in the pretreated wastes. Studies have also reported outbreaks of infections caused by *E. coli*, *S. enteritidis,* and *Y. enterocolitica*, which affect individuals at schools and prisons, increasing the medical treatment costs (94–96).

## Bacterial susceptibility to the tested antibiotics

The current study reports that *E. coli* was susceptible to doxycycline, ciprofloxacin, gentamicin, chloramphenicol, and azithromycin, and therefore, these antibiotics seem to be effective in eradicating *E. coli* infections. A similar work (97) also demonstrates that *E. coli* isolates from wastewater indicated susceptibility to ciprofloxacin (17%), gentamicin (20%), and ceftazidime (30%). Moreover, this study documents that ciprofloxacin and doxycycline are effective in killing *E. aerogenes* and *C. freundii*, hence to be considered effective in clearing *E. aerogenes* and *C. freundii* pathogenic bacteria. The results correlate with a study which reported that both *E. aerogenes* and *C. freundii* from wastewater samples were more susceptible to ciprofloxacin at 100% (38). Furthermore, *K. pneumoniae* and *K. oxytoca* had susceptibility to ciprofloxacin, gentamicin, and chloramphenicol, suggesting the efficacy of the three antibiotics. According to (98), *K. oxytoca* were susceptible to ciprofloxacin (42%). Additionally, the reported results corresponded closely to a (99) study, which detected *K. oxytoca* with 40.4% susceptibility to ciprofloxacin. In addition, P. *vulgaris* and *P. mirabilis* seem to be eliminated efficiently from the environment by use of doxycycline, ciprofloxacin, erythromycin, chloramphenicol, gentamicin, and azithromycin. This is supported by another study (100), which also detected isolates of *P. vulgaris* and *P. mirabilis* being susceptible to gentamicin (86.30%), ciprofloxacin (83.56%), chloramphenicol (78.08%), and azithromycin (38.36%). Additionally, *P. aeruginosa* was reported to be susceptible to ciprofloxacin and chloramphenicol and intermediate susceptible to ceftazidime, streptomycin, tetracycline, azithromycin, and gentamicin. This seems to suggest differential susceptibility of the bacteria species to that class of antibiotics, consistent with other studies, reporting similar isolates 100% susceptible to ciprofloxacin, 50% susceptible to vancomycin and chloramphenicol, 40% susceptible to ceftazidime, and 20.3% susceptible to gentamicin (54, 55). This work demonstrates that *S. enteritidis* loads in the general environment are efficiently reduced using ciprofloxacin, tetracycline, azithromycin, gentamicin, chloramphenicol, and metronidazole. This is compared with some results, revealing that isolates of *S. enteritis* from underground water sources conferred an increased rate of susceptibility to ciprofloxacin (100%), gentamicin (35.48%), and tetracycline (25.81%) (101). Additionally, the study reported species of *Y. enterocolitica* seem to be susceptible to ciprofloxacin, tetracycline, azithromycin, gentamicin, and chloramphenicol and intermediate susceptible to amoxicillin-clavulanic and metronidazole, implying that these antibiotics seem to be more potent in handling *Y. enterocolitica* infections globally. A similar study found *Y. enterocolitica* being susceptible to ciprofloxacin, chloramphenicol, tetracycline, gentamicin, and sulfamethoxazole-trimethoprim (102).

## Bacterial resistance to the tested antibiotics

The study reports that *E. coli* was detected with the highest percentage of multi-antibiotic resistance index of 60%, being resistant to 12 antibiotics including penicillin, ampicillin, oxacillin, ceftazidime, erythromycin, cotrimoxazole, amoxicillin-clavulanic acid, vancomycin, streptomycin, kanamycin, levofloxacin, and metronidazole. Penicillin and its antibiotic derivatives, aminoglycosidic antibiotics, and cephalosporins, that treat *E. coli* infections, seem to be reducing their strength in the treatment of *E. coli* infections. Some studies observed *E. coli* isolates being 100% resistant to ampicillin, amoxicillin-clavulanic acid, and ceftazidime (20). Other studies also detected 72% (97) and 94.7% (54) ampicillin resistance in *E. coli* from the hospital wastewater. Similarly, *E. aerogenes* and

*C. freundii* had an MDRI of 33% and 30%, respectively, being resistant to penicillin, ampicillin, amoxicillin-clavulanic acid, and sulfamethoxazole-trimethoprim. This suggests that the two bacterial species have developed resistance against the four antibiotics, rendering them ineffective in the management of infections caused by *E. aerogenes* and *C. freundii*. Resistance of *E. aerogenes* and *C. freundii* to several antibiotics is variously reported. For instance, the two bacteria species report 93.6% resistance to ceftazidime (38). Corresponding studies with the current observation state that *E. aerogenes* and *C. freundii* indicated resistance to ampicillin (98.7%), amoxicillin-clavulanic acid (85.3%) and cotrimoxazole (85.3%) (103, 104).

High resistance rates were also observed in the bacterial isolates of *K. pneumoniae* (25% MDRI) and *K. oxytoca* (30% MDRI). *K. pneumoniae* was resistant to ampicillin, amoxicillin-clavulanic acid, and ceftazidime. Furthermore, *K. oxytoca* showed resistance to penicillin, ampicillin, and sulfamethoxazole-trimethoprim. Antibiotic resistance of the two species (80%) against cephalosporins, gentamicin, ampicillin, and ceftazidime was reported (105). Moreover, both the isolates acquired 100% resistance to ampicillin, which is a similar case reported by (106). In this study, *P. mirabilis* and *P. vulgaris* demonstrated resistance against ampicillin, streptomycin, cotrimoxazole, and ceftazidime with MDRI of 25% and 17%, respectively. The results suggest varying levels of antibiotic resistance in bacteria from wastewater, which seems to be induced either by presence of antibiotic residues in the environment or heavy metals and toxic substances released by cars that are always washed alongside the River Sosiani.

In addition, *P. aeruginosa* isolates were resistant to sulfamethoxazole-trimethoprim with 13% MARI. A similar study also revealed that *P. aeruginosa* isolates were 75.41% resistant to sulfamethoxazole-trimethoprim (107). Despite the lack of antibiotic-resistant in *S. enteritis* detected in this study, a particular study still showed a high rate of antibiotic resistance in *Salmonella spp.* from the clinical samples of children to tetracycline (53%), cotrimoxazole (40%), ciprofloxacin (47%), azithromycin (27%), chloramphenicol (7%), and doxycycline (7%) (108). This seems to suggest that resistance or susceptibility of bacteria to antibiotics may be driven by different factors, such as antibiotic residues and chemicals accumulating in the environment (8, 109, 110), drug misuse, overuse, and poor hygiene, which were not investigated by the current study, but require detailed investigations to infer important insights to AMR. Additionally, the isolates of *Y. enterocolitica* conferred resistance to ampicillin, with MDRI of 13%, implying that ampicillin seems to have lost its efficacy against *Y. enterocolitica* infections; thus, the drug's strength should be boosted. A corresponding study with the current observation reports increased rates of *Y. enterocolitica* resistance to ampicillin (111).

## Conclusions

The study detected 10 pathogenic gram-negative bacterial species from River Sosiani and wastewater systems, including *E. coli, E. aerogenes, C. freundii, K. pneumoniae, K. oxytoca, P. vulgaris, P. mirabilis, P. aeruginosa, S. enteritidis,* and *Y. enterocolitica*. Among the 10 isolated bacteria, *E. coli* and *E. aerogenes* were the most prevalent bacteria contaminants detected in all the sites at frequencies of 30.3% and 20.9%, respectively. All the isolated bacteria were susceptible to doxycycline, ciprofloxacin, gentamicin, and chloramphenicol and most resistant to penicillin, ampicillin, amoxicillin-clavulanic acid, and sulfamethoxazole-trimethoprim. Furthermore, *E. coli* indicated resistance to 12 antibiotics with the highest percentage of multi-antibiotic resistance index of 60%.

## RECOMMENDATIONS

The study recommends pre-treatment of wastewater by industries, hospitals, and institutions before the release of effluents into waterbodies and regular monitoring of water sources to reduce public health risks associated with waterborne illnesses. There is a need for the County Government to enforce environment laws on wastewater disposal within the city.

## ACKNOWLEDGMENTS

The authors wish to thank University of Eldoret and the management of Eldoret Water and Sanitation Company (ELDOWAS, Kenya) for providing excellent laboratory facilities that contributed to the success of this research. We also thank the Ethical Review Committee of University of Eastern Africa Baraton for ethical review and approval of this study (Approval number: UEAB/ISER/07/03/2024), and the National Commission for Science, Technology, and Innovation for granting a research permit (NACOSTI/P/24/33905).

All authors contributed equally to the preparation and approval of the final manuscript.

## AUTHOR AFFILIATIONS

[1]Department of Chemistry and Biochemistry, University of Eldoret, Eldoret, Kenya
[2]Department of Fisheries and Aquatic Sciences, University of Eldoret, Eldoret, Kenya
[3]Department of Biochemistry and Molecular Biology, Egerton University, Njoro, Kenya
[4]Department of Biological Sciences, University of Eldoret, Eldoret, Kenya
[5]Department of Microbiology Research, Kenya Medical Research Institute, Nairobi, Kenya

## AUTHOR ORCIDs

Sharon Auma http://orcid.org/0009-0008-1993-984X
James E. Barasa http://orcid.org/0000-0001-7221-6618
Caroline Kosgei https://orcid.org/0000-0003-0392-9491
Naomi Bisem http://orcid.org/0009-0003-9488-757X
Salinah Rono http://orcid.org/0000-0001-5927-0222
Richard Korir http://orcid.org/0000-0002-1009-1140

## AUTHOR CONTRIBUTIONS

Sharon Auma, Conceptualization, Data curation, Formal analysis, Investigation, Methodology, Resources, Software, Validation, Visualization | James E. Barasa, Data curation, Formal analysis, Investigation, Methodology, Software, Supervision, Visualization | Caroline Kosgei, project administration, Validation, Visualization | Naomi Bisem, Supervision | Salinah Rono, Formal analysis, Validation | Richard Korir, Supervision

## ADDITIONAL FILES

The following material is available online.

### Open Peer Review

**PEER REVIEW HISTORY (review-history.pdf).** An accounting of the reviewer comments and feedback.

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
