## [Reviewer comments · Microbiology Spectrum]

Microbiology Spectrum

Profile of Predominant Gram-Negative Pathogenic Bacteria in River Sosiani and Wastewater Systems in Eldoret Town, Uasin Gishu County, Kenya

Sharon Auma, James Barasa, Caroline Kosgei, Naomi Bisem, Salinah Rono, and Richard Korir

Corresponding Author(s): Sharon Auma, University of Eldoret

Review Timeline:

Submission Date:	April 17, 2025
Editorial Decision:	June 8, 2025
Revision Received:	June 13, 2025
Accepted:	June 27, 2025

Editor: Blaire Steven

Reviewer(s): The reviewers have opted to remain anonymous.

Transaction Report:

DOI: <https://doi.org/10.1128/spectrum.01206-25>

Re: Spectrum01206-25 (Profile of Predominant Gram-Negative Pathogenic Bacteria in River Sosiani and Wastewater Systems in Eldoret Town, Uasin Gishu County, Kenya)

Dear Ms. Sharon Auma:

Thank you for the privilege of reviewing your work. Below you will find my comments, instructions from the Spectrum editorial office, and the reviewer comments.

After having read the manuscript and the reviewers comments I agree that this is a well written and presented manuscript. I am happy to recommend publication after a few minor comments as outlined in the attached reviewer's comments.

Revision Guidelines

Sincerely,
Blair Steven
Editor
Microbiology Spectrum

Review for Microbiology Spectrum of the manuscript 01206-25:

“Profile of Predominant Gram-Negative Pathogenic Bacteria in River Sosiani and Wastewater Systems in Eldoret Town, Uasin Gishu County, Kenya”

This study investigated the bacterial load and antibiotic resistance profiles of gram-negative bacteria in water samples from wastewater systems and River Sosiani in Eldoret town, Kenya. Pathogenic gram-negative bacteria were isolated and identified and their antibiotic susceptibility and resistance profiles against commonly used antibiotics were determined.

Introduction

Relevance and justification of the study are clearly stated in the introduction. Previous information on the topic is well described, and references are adequate and recent.

Page 5 line 89: please define previously AMR at its first use in the text (line 75).

Materials and methods

The study area, samples collection and processing, methodology in general are well described.

Results

Results are clearly presented and analyzed. Discussion is meaningful and relevant.

Line 275: include unit “mm” and revise in the entire manuscript; line 292: change to: “susceptible to ciprofloxacin”.

Discussion

Line 406: Correct to “no significant difference”.

Responses to the reviewers' comments

SECTION	REVIEWERS' COMMENT	RESPONSE
Introduction	Page 5 line 89: please define previously AMR at its first use in the text (line 75).	AMR has been defined at first before its further uses in the entire manuscript, as per the comment on line 75/76 &89 on page 5.
Results	Line 275: include unit “mm” and revise in the entire manuscript	“mm” has been included to the means and standard deviations values starting from line 275 and other sections where applicable in the entire manuscript, as commented.
	Line 292: change to: “susceptible to ciprofloxacin”	A change on “susceptible to ciprofloxacin” has been made, as commented on line 292 .
Discussion	Line 406: Correct to “no significant difference”	Correction to “no significant difference” has been made, as per the comment on line 406/409 of the manuscript file.

Re: Spectrum01206-25R1 (Profile of Predominant Gram-Negative Pathogenic Bacteria in River Sosiani and Wastewater Systems in Eldoret Town, Uasin Gishu County, Kenya)

Dear Ms. Sharon Auma:

Thank you for your prompt response to the reviewer comments. I am happy to report that I am recommending your article for publication.

Thank you for submitting your work to Spectrum.

Your manuscript has been accepted, and I am forwarding it to the ASM production staff for publication. Your paper will first be checked to make sure all elements meet the technical requirements. ASM staff will contact you if anything needs to be revised before copyediting and production can begin. Otherwise, you will be notified when your proofs are ready to be viewed.

Sincerely,
Blair Steven
Editor
Microbiology Spectrum